# Influence of Environmental Parameters and Fiber Orientation on Dissolution Kinetics of Glass Fibers in Polymer Composites

**Andrey E. Krauklis** [1,*], **Hani Amir Aouissi** [2,3,4], **Selma Bencedira** [4,5], **Juris Burlakovs** [6], **Ivar Zekker** [7], **Irina Bute** [1] and **Maris Klavins** [8]

1   Institute for Mechanics of Materials, University of Latvia, Jelgavas Street 3, LV-1004 Riga, Latvia; irina.bute@lu.lv
2   Scientific and Technical Research Center on Arid Regions (CRSTRA), Biskra 07000, Algeria; aouissi.amir@gmail.com
3   Laboratoire de Recherche et d'Etude en Aménagement et Urbanisme (LREAU), USTHB, Algiers 16000, Algeria
4   Environmental Research Center (CRE), Badji-Mokhtar Annaba University, Annaba 23000, Algeria; selmaben30@yahoo.fr
5   Laboratory of LGE, Department of Process Engineering, Faculty of Technology, Badji-Mokhtar Annaba University, Annaba 23000, Algeria
6   Chair of Rural Building and Water Management, Estonian University of Life Sciences, Kreutzwaldi 5, 51014 Tartu, Estonia; juris.burlakovs@emu.ee
7   Institute of Chemistry, University of Tartu, 14a Ravila St., 50411 Tartu, Estonia; ivar.zekker@ut.ee
8   Department of Environmental Science, University of Latvia, Jelgavas Street 1, LV-1004 Riga, Latvia; maris.klavins@lu.lv
*   Correspondence: andykrauklis@gmail.com or andrejs.krauklis@lu.lv; Tel.: +371-268-10-288

**Abstract:** Glass fibers slowly dissolve and age when exposed to water molecules. This phenomenon also occurs when glass fibers are inside fiber-reinforced composites protected by the matrix. This environmental aging results in the deterioration of the mechanical properties of the composite. In structural applications, GFRPs are continuously exposed to water environments for decades (typically, the design lifetime is around 25 years or even more). During their lifetime, these materials are affected by various temperatures, pH (acidity) levels, mechanical loads, and the synergy of these factors. The rate of the degradation process depends on the nature of the glass, sizing, fiber orientation, and environmental factors such as acidity, temperature, and mechanical stress. In this work, the degradation of typical industrial-grade R-glass fibers inside an epoxy fiber-reinforced composite was studied experimentally and computationally. A Dissolving Cylinder Zero-Order Kinetic (DCZOK) model was applied and could describe the long-term dissolution of glass composites, considering the influence of fiber orientation (hoop vs. transverse), pH (1.7, 4.0, 5.7, 7.0, and 10.0), and temperature (20, 40, 60, and 80 °C). The limitations of the DCZOK model and the effects of sizing protection, the accumulation of degradation products inside the composite, and water availability were investigated. Dissolution was experimentally measured using ICP-MS. As in the case of the fibers, for GFRPs, the temperature showed an Arrhenius-type influence on the kinetics, increasing the rate of dissolution exponentially with increasing temperature. Similar to fibers, GFRPs showed a hyperbolic dependence on pH. The model was able to capture all of these effects, and the limitations were addressed. The significance of the study is the contribution to a better understanding of mass loss and dissolution modeling in GFRPs, which is linked to the deterioration of the mechanical properties of GFRPs. This link should be further investigated experimentally and computationally.

**Keywords:** glass fibers; composites; environmental aging; modeling; kinetics; water; pH; temperature; orientation; durability

## 1. Introduction

The most commonly used type of fiber reinforcement is glass fiber (GF) [1–3]. GFs are hydrophilic and are susceptible to degradation; they dissolve due to environmental aging when exposed to water molecules when submerged in liquid media or humid environments [4]. The fact that GFs degrade in aqueous environments has been known at least since the early 1970s [5–7]. However, not all GFs degrade at the same rates [8]. Furthermore, this process, albeit slower, also occurs when GFs are encapsulated in composites—glass fiber-reinforced polymers (GFRPs) [9]. The encapsulated GFs are protected by the sizing and surrounding polymer matrix [10]. Dissolution rates are lower for protected fibers in GFRPs than for single fibers or fiber bundles. However, this protection is insufficient to stop the degradation process altogether [10]. Such environmental aging results in the deterioration of the composites' mechanical properties, negatively affecting the strength and modulus of the GFRPs [11]. The negative effect can be so strong that it cannot be neglected when designing structural GFRPs for use underwater and in humid environments [12].

The typical design lifetime of GFRP structures in structural applications can range from 25 to 40 years or even more [13,14]. During this period, the GFRPs may be continuously exposed to water molecules, leading to environmental degradation of the reinforcing GFs [12–14].

**Glass formulation.** The rate of the degradation process depends on the nature of the glass material formulation (E, ECR, A, R, S-glass, etc.), the sizing formulation, the fiber orientation, and environmental factors such as acidity (pH), temperature, and mechanical stress [15,16]. The degradation rate may vary from extremely slow to extremely fast, depending on the material properties and environment, i.e., type of acid and pH. According to [17], the degradation of GFRPs due to the environment's pH mostly depends on the type of glass fibers used. For example, boron-free glass fibers of the ECR type are considered the most inert, whereas our tests on R-glass have shown that R-glass fibers degrade relatively fast in strongly acidic environments. GFRP pipes with ECR glass fibers are often used in strongly acidic applications, where they can withstand degradation for decades. ECR stands for E-glass Corrosion Resistant. It should not be confused with the most common E-glass, which degrades rather quickly when exposed to strong acid like the R-glass does.

It was found that the corrosion resistance of the glass fibers varies enormously among glass fiber types. Furthermore, the results indicate that the laminates' stress-corrosion properties correlate with the fibers' uniform corrosion resistance. The performance and reliability of an FRP structure exposed to aggressive environments can be strongly improved by choosing a corrosion-resistant fiber. The risk of stress-corrosion cracking can be reduced to a minimum level [18].

In this work, the degradation process of typical industrial-grade R-glass fibers (R-GFs) inside an epoxy fiber-reinforced composite was studied experimentally and computationally.

**Glass fiber dissolution kinetics determination.** Current recycling strategies have attempted to mitigate the environmental harm caused by the disposal of end-of-life composite materials, some of which are currently being employed on an industrial basis [1]. Understanding and managing the dissolving process of glass fiber composites necessitate research into reaction kinetics. The dissolution of silicate glass within alkali solutions and environments has been well described [19], whereas the glass fiber dissolution mechanism and kinetics are less documented (see the following chapters for more detail). The reaction rate constant for glass fiber dissolution in alkaline solution at 95 °C has been found to be $1.3 \times 10^{-4}$–$4.3 \times 10^{-4}$ g/(m$^2$ s). The reaction order ($n$) is 0.31–0.49 in alkaline solution, with the activation energy being 58–79 kJ/mol [20].

Various studies have been carried out on the application of dissolution and elemental measurements of glass fibers, casework glass, headlamps in automotive applications [21], and glass in forensic science [22] with the use of mass spectrometric, spectroscopic, and radiochemical techniques [23]. In one study, the authors reacted E-glass fibers with a corrosive medium, removing residues, and found changes in the fiber dimensions using a scanning

electron microscope (SEM) [20]. Among spectrometric and mass spectrometric methods, glass material composition has been characterized by techniques such as laser ablation inductively coupled plasma mass spectrometry (LA-ICP-MS) [24] and ICP-MS (especially suitable for multi-elemental determination), such as in [10,16,21,25,26]. LA-ICP-MS has an advantage over ICP-MS because there is no need for sample dilution prior to analysis in the former; however, the technique was not feasible, and therefore, ICP-MS was used in this study. In [27], a technique for determining the dissolution rate constant of a borosilicate glass fiber in the lung, as determined in vitro, from the oxide composition in weight percent was described. Some other methods used for glass elemental characterization include atomic X-ray fluorescence [22], atomic absorption spectroscopy [24,28], SEM [29], and neutron activation [30]. Composite measurements by thermogravimetric analysis (TGA) can define the moisture, fiber content, and polymer content in GFRPs. Micro-computed tomography (μCT) has been applied to determine voids in glass and the fiber bundle geometry in polymers relying on fiber-reinforced glass (GFR) materials [31]. Optical coherence tomography (OCT), as a nondestructive method, has been used for GFR material determination [32], whereas X-ray CT surpasses the former technique by displaying a clearer reinforcement structure [31].

**Environmental factors and orientation effect.** The kinetics of the aging process depends on environmental factors such as temperature, acidity levels, and mechanical loads [16]. Previous studies have investigated these environmental effects experimentally and computationally for the same R-GFs (fiber bundles, not encapsulated into composites) [16,27]. Furthermore, the aging process also depends on the layup and orientation of the reinforcement GFs [10]. It involves composite-specific response effects, such as sizing (also for sized GFs [10]) and matrix protection, the accumulation of degradation products inside the composite, and water availability [10]. These effects have been studied for R-glass GFRPs with identical layups and orientations to those in the work in [10], but only in neutral conditions and without the influence of temperature. Hence, in this work, the synergy was investigated, proceeding from findings on the fiber orientation effects studied in [10] and the environmental impact on R-GF bundle aging studied in [16].

However, it should also be noted that the polymer matrix protects the glass in composites and should slow the dissolution. There have been no conclusive results on how much the matrix slows the dissolution, except for a single study, where it was found that even for thin composite plates, dissolution slows down about twofold when glass fibers are embedded in a polymer [10]. Additionally, it was found that the fiber orientation affects the glass dissolution rate. The dissolution-driven degradation of GFRPs with fibers in a hoop orientation was slower than for plates with a transverse fiber orientation [10].

**The molecular mechanism of R-GF degradation.** Glass fiber dissolution-driven degradation occurs when silica glass material (GF) is in contact with water molecules. A respective hydrolysis reaction takes place and is shown in Figure 1, according to [33].

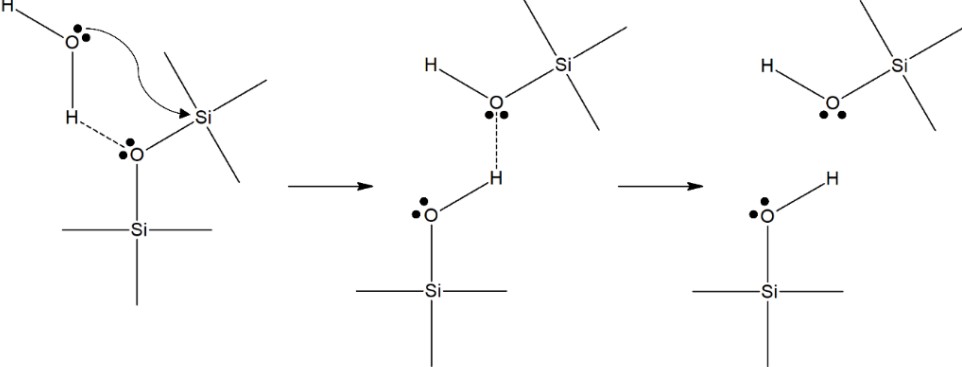

**Figure 1.** Hydrolysis chemical reaction of silica glass network, according to [33].

Most studies explain the environmental degradation mechanisms of glass in terms of surface reactions, chemical affinity, and diffusion [4,34–38]. However, dissolution experiments have been mainly reported for bulk glass, whereas GFs and GFRPs have rarely been studied [27]. However, some reflections on the mechanism of GF degradation do exist. The state-of-the-art mechanistic understanding of GF degradation is based on the works by Grambow et al. (2001), Hunter et al. (2015), and Echtermeyer and Krauklis et al. (2018 and 2019) [11,16,27,39,40]. The complex nature of GF degradation involves several parallel processes, namely, gel layer formation, dissolution of glass matrix constituents, alkaline and alkaline earth ion exchange, and neoformation of solid reaction products. Some of these reactions occur in the glassy state, while others lead to the leaching of the reaction products into the surrounding aqueous environment [39,41]. Competing chemical reactions can be described by Equations (1)–(12), summarized from sources from 1979 up to 2022 [6,8,15,16,27,33,39–44]:

$$(\equiv Si - ONa) + H_2O \rightarrow (\equiv Si - OH) + OH^- + Na^+ \tag{1}$$

$$(\equiv Si - OK) + H_2O \rightarrow (\equiv Si - OH) + OH^- + K^+ \tag{2}$$

$$(\equiv Si - O)_2Ca + H_2O \rightarrow 2(\equiv Si - OH) + 2OH^- + Ca^{2+} \tag{3}$$

$$(\equiv Si - O)_2Mg + H_2O \rightarrow 2(\equiv Si - OH) + 2OH^- + Mg^{2+} \tag{4}$$

$$(\equiv Si - O - Al =) + H_2O \leftrightarrow (\equiv Si - OH) + (= Al - OH) \tag{5}$$

$$(\equiv Si - O)_2Fe + H_2O \rightarrow 2(\equiv Si - OH) + 2OH^- + Fe^{2+} \tag{6}$$

$$(\equiv Si - O)_3Fe + H_2O \rightarrow 3(\equiv Si - OH) + 3OH^- + Fe^{3+} \tag{7}$$

$$(\equiv Si - O - Si \equiv) + OH^- \leftrightarrow (\equiv Si - OH) + (\equiv Si - O^-) \tag{8}$$

$$(\equiv Si - O^-) + H_2O \leftrightarrow (\equiv Si - OH) + OH^- \tag{9}$$

$$SiO_2 + H_2O \leftrightarrow H_2SiO_3 \tag{10}$$

$$H_2SiO_3 + H_2O \leftrightarrow H_4SiO_4 \tag{11}$$

$$MeCl_x \xrightarrow{H_2O} (Me^{x+}) + xCl^- \tag{12}$$

The degradation of GFs proceeds in two phases. In the short-term non-steady state (Phase I), hydrolytic degradation involves competing processes (ion exchange, gel formation, and dissolution). In the long-term steady-state (Phase II), hydrolytic degradation is governed by the glass dissolution mechanism and follows zero-order reaction kinetics [9,10,27]. Such kinetics depends on the glass surface area in contact with water, proportional to the fiber radius. As the dissolution continues, the radius decreases, resulting in mass loss deceleration [9]. For the studied R-glass, the transition from Phase I into Phase II occurs in about a week (166 h) at pH 5.7 and 60 °C [9,10,27]. The elements that are released during the degradation of R-glass are Na, K, Ca, Mg, Fe, Al, Si, and Cl [27]. The experimental glass mass loss (measured by ICP-MS) is the cumulative mass loss of all of these ions [27]. The Si contribution to the total mass loss of the studied R-glass is the largest (56.1 wt. %) and seems to govern the dissolution process [27].

Dissolution-driven degradation is an energy-activated process. It is known that for non-embedded R-GFs, the process follows the Arrhenius principle well: the rate of dissolution increases as the temperature increases [16]. The temperature dependence of a dissolution rate constant can be described using the Arrhenius equation (Equation (13)), being an exponential function:

$$K_0 = Ae^{-\frac{E_A(pH,\sigma)}{RT}} \tag{13}$$

where $A$ is the pre-exponential factor (g/(m²·s)); $R$ is the gas constant, with a value of 8.314 J/(mol·K); $T$ is the absolute temperature (K); and $E_A$ is the activation energy (J/mol). Both pH and stress corrosion affect the activation energy term in the Arrhenius equation [16].

Mechanical stress accelerates the rate due to a stress-corrosion mechanism [16]. However, the most prominent environmental influence on kinetics is due to the parabolic pH influence. The dissolution rate is slowest in conditions close to neutral and accelerates towards both acidic and basic ends, especially in highly acidic environments [16].

In this work, the dissolution kinetics was measured experimentally using ICP-MS. A Dissolving Cylinder Zero-Order Kinetic (DCZOK) model was then applied to investigate the long-term dissolution of glass composites computationally, considering the influence of fiber orientation, pH, and temperature. This model was chosen for the calculations because it considers the complex short-term and dissolution-dominated long-term processes, describes dissolution, and is able to link it to the radius reduction kinetics, crack-growth kinetics, and strength reduction kinetics [11]. Another aspect of the choice of calculation model was the ability to avoid introducing additional terms, such as a conversion factor [27]. The DCZOK model is described in more detail in [16,27], similar to the work performed in the two studies mentioned above involving neutral environment effects on GFRPs [10] and environmental effects on R-GF [16]. The total material loss and release of Si under various environmental conditions were simulated using the DCZOK model, and rate constants were obtained and reported.

**The benefit for the industry.** The industry is interested in pH, temperature, and stress corrosion as environmental effects [16]. Temperatures are also attractive to the industry for accelerated testing purposes [12]. As the industry is concerned with lowering the testing time for fiber-dominated property deterioration in GFRPs, the model considerably shortens the experimental testing time to the short term and slightly enters the long-term steady state to obtain model parameters, i.e., kinetic constants and time to reach the steady state. While it takes time to obtain the parameters, the most significant time saving comes from using kinetic constants and the model [12,15,45].

Along with the current technological limitations of composite recycling (although the technology is rapidly developing) [1], the environmental durability of GFRPs remains one of the limiting factors in the development of the composite industry [12,46]. This is due to the superior mechanical properties being compromised by the uncertainty of the material's interaction with the environment [47]. Modeling can address these questions and solve the problem at hand [12].

Among the state-of-the-art literature, an analytical modeling toolbox (Modular Paradigm for GFRPs) consisting of seven modules for reinforcement materials was described in [8]. The work in [8] states that the extension of the DCZOK model to GFRPs is currently a major challenge. Therefore, this study contributes to a better understanding of modeling the mass loss of glass fibers in GFRPs (Module 4 in [8]).

**This study aimed** to experimentally characterize the environmental aging of R-GFRPs in various fiber orientations and environmental conditions and to capture this behavior computationally using the DCZOK model. The secondary aim is to investigate the limitations of the DCZOK model and better understand the effects of sizing protection, the accumulation of degradation products inside the composite, and water availability.

## 2. Materials and Methods

### 2.1. Materials

**Glass fibers.** Common industrial-grade boron-free and fluorine-free 3B HiPer-Tex W2020 R-GF was studied as a reinforcement material (in the form of stitch-bonded fabrics). 3B HiPer-Tex W2020 R-glass is classified as a high-strength and high-modulus R-glass per the definition by an international standard [48]. The average fiber diameter was $17 \pm 2$ μm, and the density was 2.54 g/cm$^3$, as reported in previous experimental and modeling studies on the same material [16,49]. The authors estimated 4098 fibers per bundle in a mat on average [16,49]. The specific surface area was 0.09 m$^2$/g based on geometrical considerations (as a product of the number, circumference, and length of the fibers [27]) and 0.18 m$^2$/g according to Brunauer–Emmett–Teller (BET) tests; the difference is explained by

uneven sizing distribution [26]. All reinforcement within this study had the same W2020 sizing (typical industrial epoxy-compatible sizing) as in previous studies [9–11,16,26,27].

The dissolution of the same unsized R-glass fibers was previously studied in other works by Krauklis and Echtermeyer [9–11,16,26,27]. In addition, GFRPs with the same fibers and sizing but limited environmental conditions were studied in [10]. This work is a continuation of these studies and an investigation of synergy.

**Polymer composites.** Composites were prepared using a vacuum-assisted resin transfer molding (VARTM) process in two unidirectional configurations (fiber orientations): in-plane (hoop orientation when cut from a pipe) and out-of-plane (transverse direction when cut from a pipe); see Figure 2 for a clear visual.

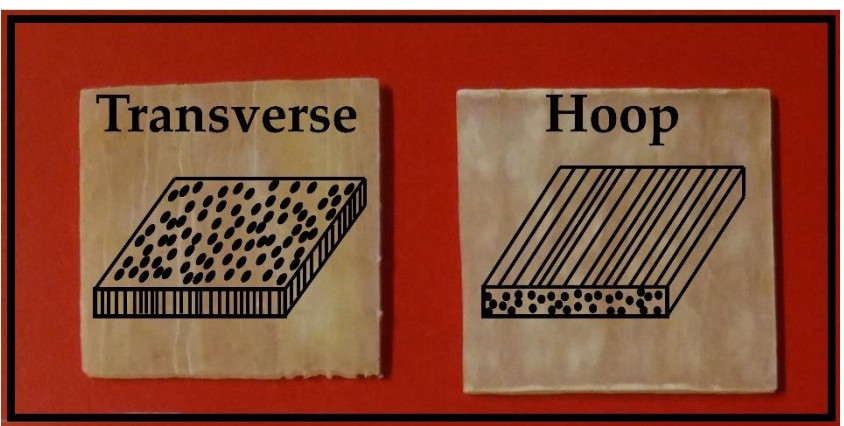

**Figure 2.** GFRP plate specimen fiber orientations: (**left**) transverse and (**right**) hoop orientations.

Composite laminates were prepared via vacuum-assisted resin transfer molding (VARTM) using Hexion epoxy resin RIMR135 and amine hardener RIMH137 in a stoichiometric proportion of 100:30 (weight ratio). The resin and the curing agent consisted of bisphenol A diglycidyl ether (DGEBA), 1,6-hexanediol diglycidyl ether (HDDGE), poly (oxypropylene) diamine (POPA), and isophorondiamine (IPDA). Before VARTM, the epoxy/hardener mixture was degassed in a vacuum chamber for 30 min to remove bubbles. Curing was performed at room temperature for 24 h, followed by post-curing in an air oven at 80 °C for 16 h. The composite laminate was cut into rectangular bars and subsequently into composite plates with dimensions of 20 mm × 20 mm × 1.5 mm and with fibers oriented parallel (hoop-orientation plates) or normal (transverse-orientation plates) to the large face of the plate (as shown in Figure 2). The specified dimensions were achieved within 5 percent tolerance.

The specimens were dried for two weeks after the preparation procedures until they reached equilibrium. Water content was monitored via the gravimetric method and a Fourier transform infrared (FTIR) spectroscopy composite water monitoring method, described in [50]. FTIR spectra were recorded using Varian Scimitar 800 FT-IR in Attenuated Total Reflectance (ATR) mode via Pike technologies GladiATR$^{TM}$ mode. Spectra were obtained at 4 cm$^{-1}$ resolution, with 50 scans co-added over a range of wavenumbers from 400 to 4000 cm$^{-1}$.

GFRP plate specimen configurations are summarized in Table 1.

**Aging medium.** Distilled water (0.5–1.0 MΩ·cm) was used to condition the R-GFRP specimens. The pH of the distilled water was 5.65 ± 0.01, which is lower than neutral due to dissolved $CO_2$ from the atmosphere in equilibrium. IUPAC standard buffer solutions (Radiometer analytical, France) were used to study the pH effect on the kinetics of R-GFRP dissolution. Solutions of pH 1.679, 4.005, 5.650, 7.000, and 10.012 were used. The pH values of the solutions were checked using a standard pH-meter (MeterLab PHM210) with an accuracy of ±0.01 pH. All specimens were dry when placed in the water solutions, meaning they were all saturated at their respective pH and temperature.

**Table 1.** GFRP plate specimen configurations.

| Geometry [mm] | 20 × 20 × 1.5 |
|---|---|
| Loss on ignition (LOI) | 0.0064 |
| Composite density [g/cm$^3$] | 1.93 |
| Glass density [g/cm$^3$] | 2.54 |
| Polymer density [g/cm$^3$] | 1.1 |
| Sizing density [g/cm$^3$] | 1.1 |
| Fiber volume fraction | 0.5950 |
| Void volume fraction | 0.0044 |
| Polymer volume fraction | 0.3920 |
| Sizing/interphase volume fraction | 0.0087 |
| Fiber mass fraction | 0.7723 |
| Polymer mass fraction | 0.2227 |
| Sizing/interphase mass fraction | 0.0049 |

*2.2. Methods*

**Glass fiber fraction determination.** The fractions of the glass fibers of the GFRP composite specimens were determined by the burn-off test according to the ASTM Standard D3171 and used in density measurements [51]. The densities of the matrix polymer ($\rho_{matrix}$) and glass fiber ($\rho_{glass}$) were 1.1 g/cm$^3$ and 2.54 g/cm$^3$, respectively. The density of the composite ($\rho_{GFRP}$) was determined to be 1.93 g/cm$^3$ by measuring the mass and dimensions of a large GFRP composite block. The volume and mass fractions of the matrix polymer can then be calculated using Equations (14) and (15), respectively.

$$V_f = \frac{\rho_{GFRP} - \rho_{matrix}}{\rho_{glass} - \rho_{matrix}} \tag{14}$$

$$m_f = \frac{\rho_{glass} \cdot V_f}{\rho_{matrix} \cdot \left(1 - V_f\right) + \rho_{glass} \cdot V_f} \tag{15}$$

The volume and mass fraction of fibers were $V_f$ = 0.595 and $m_f$ = 0.772, respectively. The void content was very low (0.44%), similar to [52].

**Diffusion measurements and environmental aging.** Dissolution experiments in the water of GFRP specimens were conducted using a batch system. Specimens for the dissolution study were weighed using analytical scales (AG204, with a precision of ±0.1 mg) before and during the experiments. The specimens were placed in inert closed vessels filled with 50 mL of distilled water or pH buffer solutions. The tight sealing of specimens was ensured. The water-tight containers containing specimens and water solutions were placed in a water bath. Gravimetric water uptake was measured for GFRP plates in hoop and transverse orientations at pH 5.65 at 60 °C to ensure that saturation with water was reached. The bath's water temperature (20, 40, 60, and 80 °C) was controlled via PID-controlled heating, giving an accuracy of ±1 °C. The two-stage heating system was used to ensure that there was no contact of the sample water with other potential ion-releasing sources, such as the heating element itself.

**Microscopy of aged specimens.** Optical microscopy was performed using a digital microscope (Hirox RH-2000) equipped with an MXB-2500REZ lens with a magnification of 140 and resolution of 1.06 μm. Microscopy was used to inspect changes in GFRP structure and morphology after exposure to various environmental aging conditions.

**Glass material dissolution (ion release) measurements.** The concentration of dissolved ions was analyzed over time via inductively coupled plasma mass spectrometry

(ICP-MS) to obtain glass material dissolution kinetics experimentally. The total mass loss of glass material was measured as the sum of all ions' release quantified by ICP-MS (cumulatively over time). Analyses were performed using a double-focusing magnetic sector field ICP-MS (Finnigan ELEMENT 2, Thermo-Scientific, Waltham, MA, USA), equipped with a sample introduction system (PrepFAST, ESI/Elemental Scientific, Omaha, NE, USA) and a pre-treatment/digestion UltraClave (Milestone). Acidification of samples was performed using ultra-pure grade $HNO_3$ SubPur (Milestone) to prevent the adsorption of ions to the wall of the sample vials. Experiments were performed with three parallels.

The benefit of ICP-MS versus gravimetric analysis is that it allows for measuring the dissolution kinetics of each separate ion, as well as the total mass loss [27]. In addition, it allows the decoupling of inorganic material degradation, such as glass, from the organic polymer. The data obtained from the ICP-MS experiments were in the form of mass concentration at each time point (non-cumulative) c (g/L) and were converted to the $m_{dissolved}$ form by using Equation (16):

$$m_{dissolved} = V_{water} \int_0^t c\,dt \tag{16}$$

where $V_{water}$ is the volume of a water sample in the ICP-MS measurement. The $V_{water}$ used for experiments was 50 mL. Equation (3) is valid for each ion release and the total mass loss.

**The Dissolving Cylinder Zero-Order Kinetics (DCZOK) Model.** The analytical DCZOK model can predict the mass loss kinetics, fiber radius reduction kinetics, hydrolytic flaw growth kinetics, and hydrolysis-induced strength degradation kinetics of unembedded R-GFs [9,11]. The dissolution of GFs inside composites is slower compared to GF bundles and is addressed in the analytical model [10].

For fibers in infinite water availability conditions, the dissolution, which is a surface reaction, can be well-described with zero-order kinetics [27]. However, the decrease in fiber radius and thus the decrease in surface area with time should be accounted for [27]. For sized fibers, the effect of sizing $\xi_{sizing}$ should also be accounted for [10]. The effect of sizing on glass dissolution $\xi_{sizing}$ for the studied R-glass is 0.165, protecting fibers from water by almost an order of magnitude in terms of the dissolution rate [10]. For sized fiber bundles (not embedded in the composite), the mass loss kinetic model equation in differential form is (Equation (17)):

$$\frac{\partial m}{\partial t} = 2n\pi l \left( r_0 K_0 \xi_{sizing} - \frac{K_0^2 \xi_{sizing}^2}{\rho_{glass}} t \right) \tag{17}$$

The radius reduction over time is accounted for in the model; the environmental parameters and their synergy, such as pH, temperature, and stress corrosion, affect the material–environment energy-activated interactions, thus affecting the dissolution rate constant $K_0$ [9]. For unembedded R-GFs, the DCZOK model was successfully applied to account for the environmental conditions (pH, $T$, and $\sigma$), as shown in Equation (18) [16]:

$$\frac{\partial m}{\partial t} = 2n\pi l \left( r_0 A e^{-\frac{E_A(pH,\sigma)}{RT}} \xi_{sizing} - \frac{\left( A e^{-\frac{E_A(pH,\sigma)}{RT}} \xi_{sizing} \right)^2}{\rho_{glass}} t \right) \tag{18}$$

where $m$ is the total cumulative mass dissolved after time $t$; $K_0$ is the material–environment interaction property; $\xi_{sizing}$ is the protective effect of sizing; $pH$ is the acidity of the environment $(-)$, $T$ is its temperature (K); $\sigma$ is mechanical stress (MPa); $n$ is the number of fibers $(-)$; $l$ is the length of fibers (m); $r_0$ is the initial fiber radius (m); $\rho_{glass}$ is the density of glass (g/m$^3$); $A$ is the pre-exponential factor (g/(m$^2$·s)); $R$ is the gas constant with a value of 8.314 J/(mol·K); $T$ is the absolute temperature (K); and $E_A$ is the activation energy (J/mol).

For GFRPs, an extended version of the DCZOK model is required, as discussed in [9,10]. As aging advances, the degradation products accumulate inside the composite material

and subsequently slow the reaction rate of glass fiber dissolution by shifting the chemical equilibrium. Since the long-term response is governed by Si dissolution [27], the silica hydrolysis products cause the deceleration of glass dissolution inside the composites [9]. In the model, the accumulation term accounts for a "driving force" term, which shows that the mass-loss rate is proportional to the difference between concentrations of degradation products inside the composite at saturation and at a specific time point [10]. The extended DCZOK model can be mathematically expressed as Equations (19) and (20), and the model considering environmental effects is described by Equation (13) [9,10,16]:

$$\frac{\partial m}{\partial t} = K_0 \xi_{sizing} S C_{H_2O}^{n_{order}} \left( C_{SiO_2}^{eq} - C_{SiO_2} \right)^{m_{order}} \cong K_0^* S \tag{19}$$

$$\frac{\partial m}{\partial t} = 2n\pi l \left( r_0 K_0 \xi_{sizing} C_{H_2O}^{n_{order}}(t) \left( \frac{C_{SiO_2}^{eq} - C_{SiO_2}(t)}{C_{SiO_2}^{eq}} \right)^{m_{order}} - \frac{\left( K_0 \xi_{sizing} C_{H_2O}^{n_{order}}(t) \left( \frac{C_{SiO_2}^{eq} - C_{SiO_2}(t)}{C_{SiO_2}^{eq}} \right)^{m_{order}} \right)^2}{\rho_{glass}} t \right) \tag{20}$$

where $m$ is a total cumulative mass dissolved after time $t$; $K_0^*$ is an apparent reaction kinetic constant that can be obtained from the regression of experimental data. While $K_0$ is a material property, $K_0^*$ incorporates the effects of sizing, water availability, and degradation product accumulation; $n$ is the number of fibers (−); $l$ is the length of fibers (m); $r_0$ is the initial fiber radius (m); $\rho_{glass}$ is the density of glass (g/m$^3$); $\xi_{sizing}$ is the protective effect of the sizing; $S$ is the glass surface area exposed to water; $C_{H_2O}$ is the availability of water molecules to the reacting glass surface; $n_{order}$ is the order of the reaction; $C_{SiO_2}^{eq}$ is the concentration of degradation products at saturation inside the composite; $C_{SiO_2}$ is the current concentration of degradation products inside the composite; and $m_{order}$ is the order of the driving force term.

**Assumptions.** The model involves the following assumptions. As a simplification, this model is deterministic, and all fibers are assumed to have the same initial radius $r_0$; the cross-sectional surface area at the end of the fibers is assumed to be negligible in the calculation of the surface area; the length of the long fibers $l$ is assumed to be constant during the whole dissolution process. During the whole degradation process, the density of the glass material is assumed to be constant $\rho_{glass}$. The effect of sizing $\xi_{sizing}$ is assumed to be independent of environmental conditions and time [9,10]. For the studied R-GFRPs, the protective effect of sizing $\xi_{sizing}$ (0.165) on glass fiber dissolution was found to be about six times [10]. For free fiber bundles (not embedded in the composite), the conditions of infinite availability of water are ensured by using large volumes of water, thus making the rate of reaction independent of the water concentration [10]. For composites, as aging proceeds, degradation products accumulate inside the composite plates and slow the rate of the reaction. Since the long-term reaction is governed by Si dissolution [27], the silica hydrolysis products are what cause the deceleration of glass dissolution inside the composites. In the model, the accumulation term is accounted for as a driving force term that shows that rate of mass loss is proportional to the difference between the saturation ($C_{SiO_2}^{eq}$) and current concentrations ($C_{SiO_2}$) of degradation products in the composite and the order ($m_{order}$). The order of water availability $n_{order}$ accounts for the effect of the state of the water present in the polymer surrounding the glass, i.e., in a free, bound, or mixed state; thus, if eclectic, it can be a fractional number. The closer $n_{order}$ is to 0, the more water there is in the free state. For free fiber bundles (not embedded in the composite) in a large volume of water, conditions of infinite availability of water ensue, the rate of the reaction becomes independent of the water concentration, and the water availability order $n_{order}$ becomes 0 [9,10].

## 3. Results

**Microscopy.** The authors performed a short experimental study earlier by immersing thin R-glass/epoxy GFRP plates (with two fiber orientations, hoop and transverse) at various pH values to see whether there was significant damage due to the short-term hydrolysis of glass fibers in GFRPs. GFRPs were immersed in aqueous solutions of pH 1.7, 4.0, 5.65, 7.0, and 10.0. The results are shown in Figure 3.

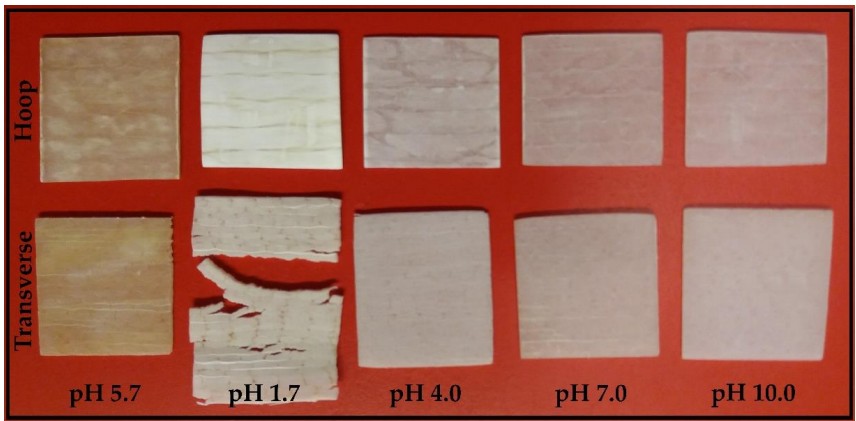

**Figure 3.** Effect of various pH levels of aqueous solutions on the degradation of R-glass GFRP plates with two fiber orientations (hoop and transverse orientations) after about a week (pH 1.7, 4.0, 7.0, and 10.0) and after four months (pH 5.7) of exposure.

Composite plates with two fiber orientations (hoop and transverse) are shown in Figure 3, and microscopy images are shown in Figures 4–11. GFRP plates were exposed to aqueous solutions of various pH values for a few days at room temperature. The composite specimens quickly degraded at pH 1.7 and were destroyed within a week, as seen in Figure 3. Thus, GFRPs should not be used in strongly acidic conditions. This observation is in full accord with a finding in another study that stated that many GFRPs fail catastrophically after a critical time of exposure to acids [17]. Furthermore, it can be seen that failure occurred on the surfaces of fibers, meaning that it was due to the failure of either glass fibers or the fiber–matrix interphase, which agrees with [26].

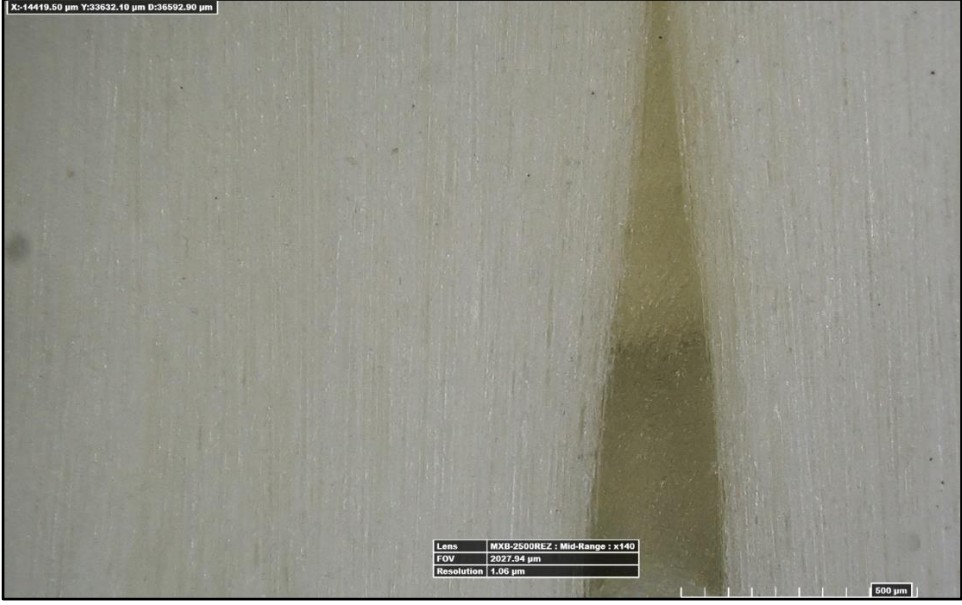

**Figure 4.** *Cont.*

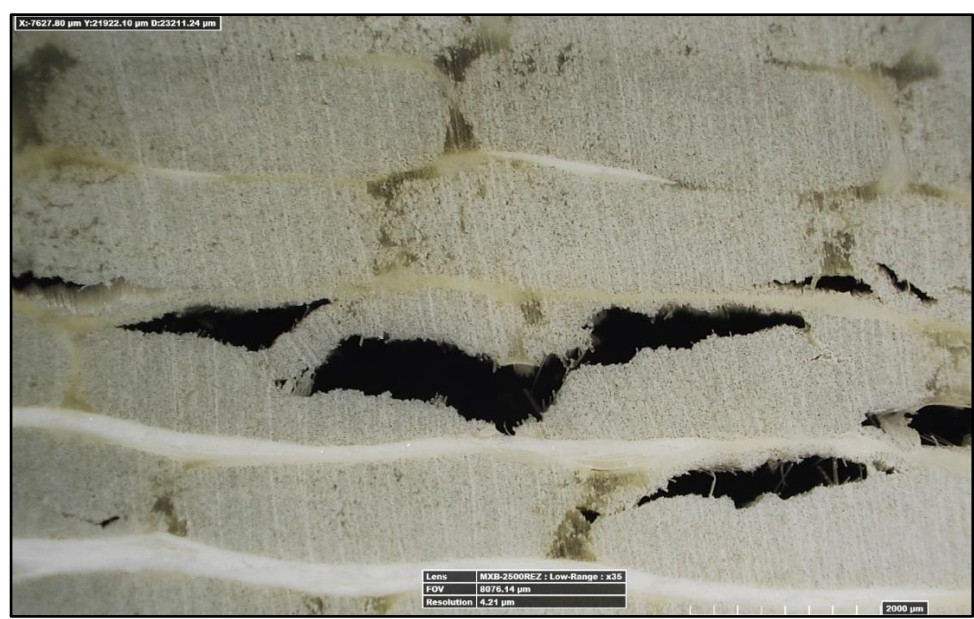

**Figure 4.** R-glass FRP plate with a (**bottom**; ×35 magnification) transverse and (**top**; ×140 magnification) hoop fiber orientation after a week of exposure to pH 1.7 aqueous solution.

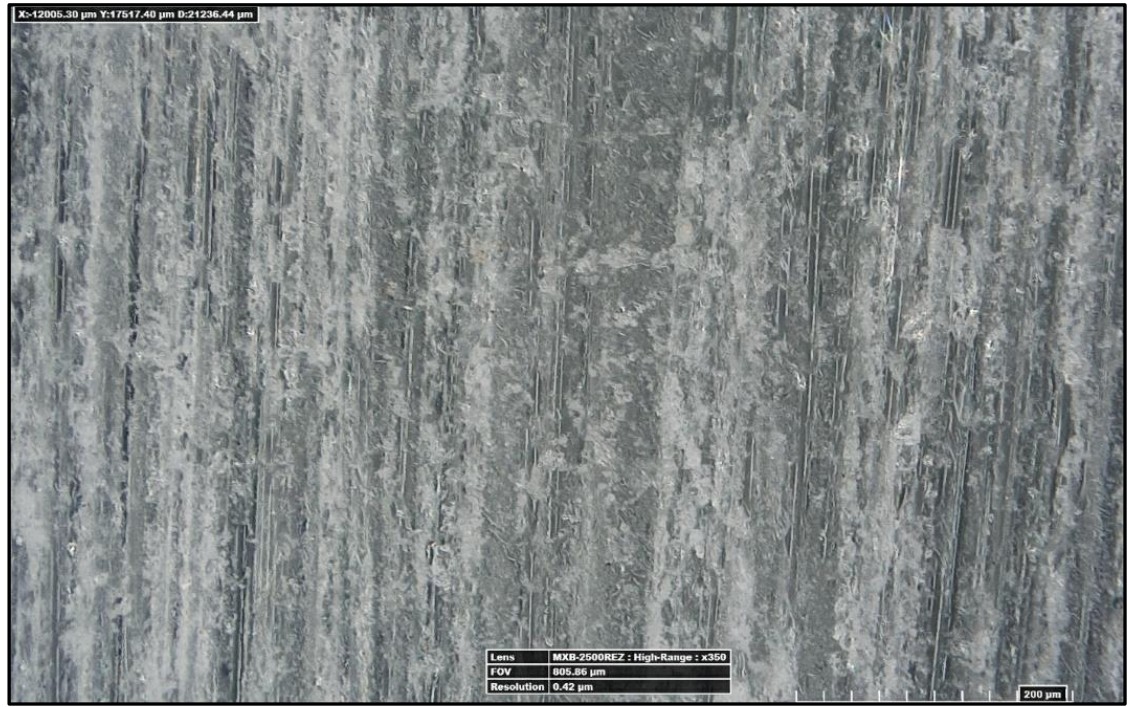

**Figure 5.** *Cont.*

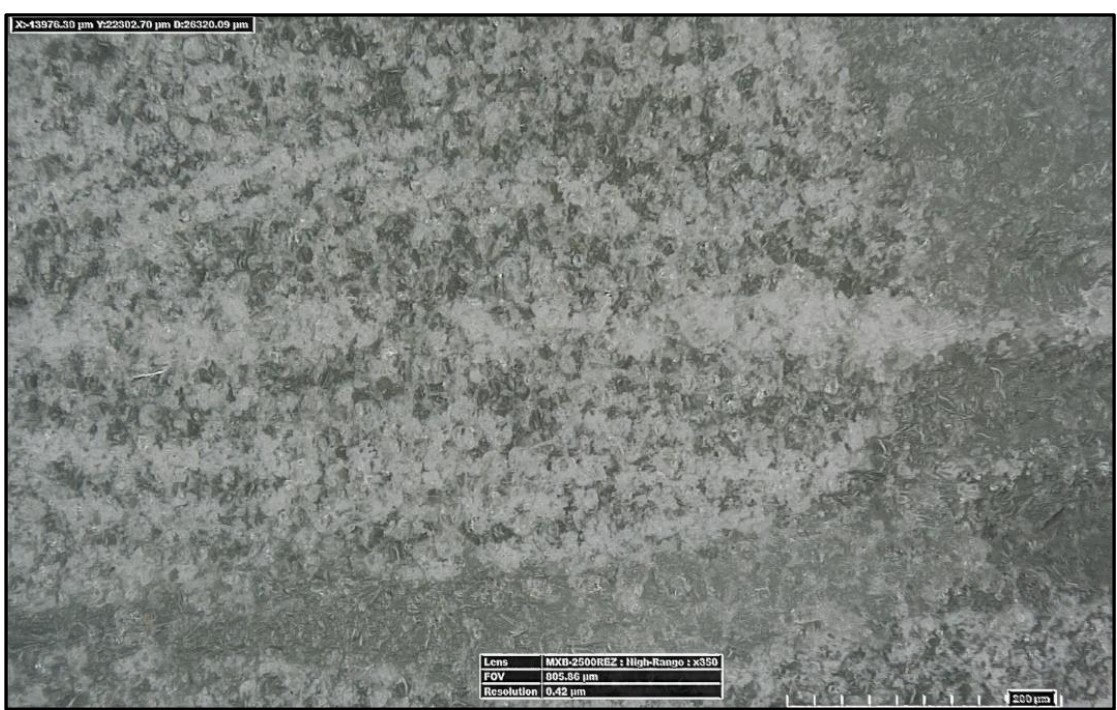

**Figure 5.** R-glass FRP plate with a (**bottom**; ×350 magnification) transverse and (**top**; ×350 magnification) hoop fiber orientation after a week of exposure to pH 4.0 aqueous solution.

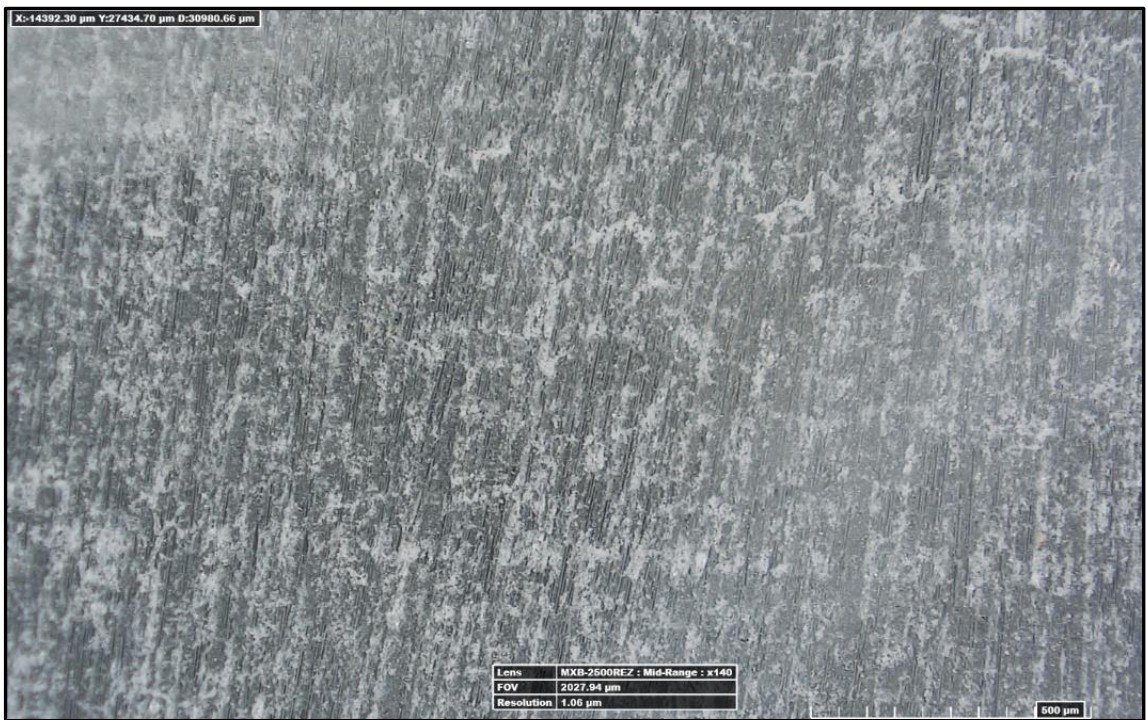

**Figure 6.** *Cont.*

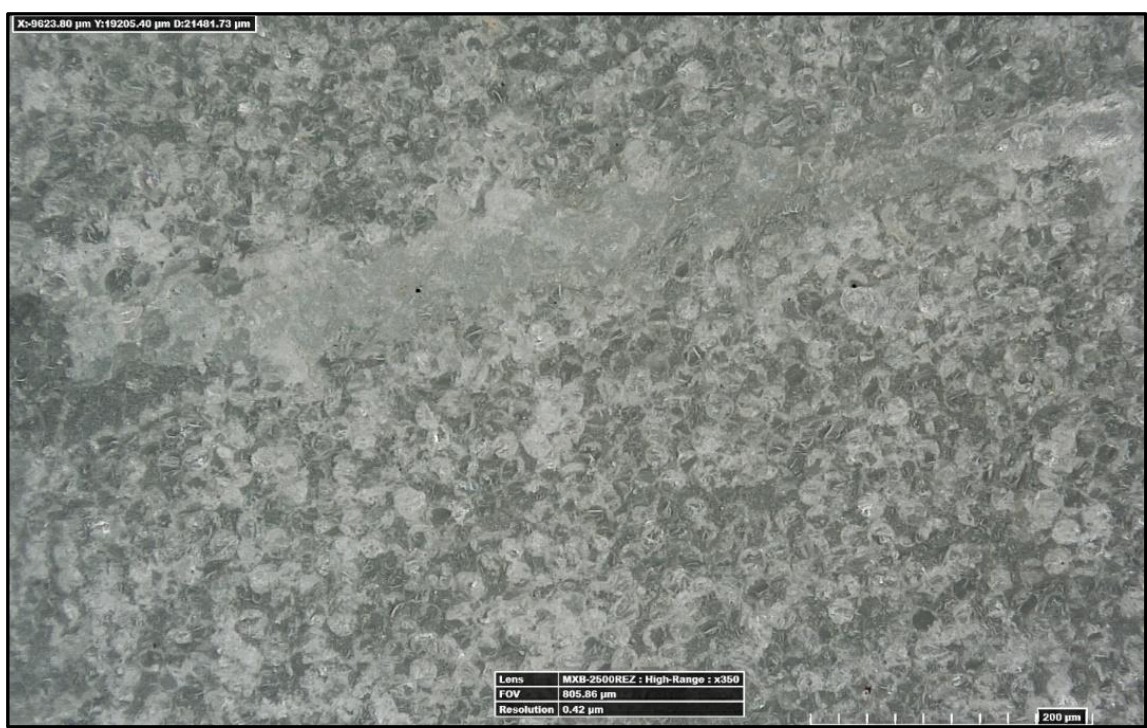

**Figure 6.** R-glass FRP plate with a (**bottom**; ×350 magnification) transverse and (**top**; ×140 magnification) hoop fiber orientation after a week of exposure to pH 7.0 aqueous solution.

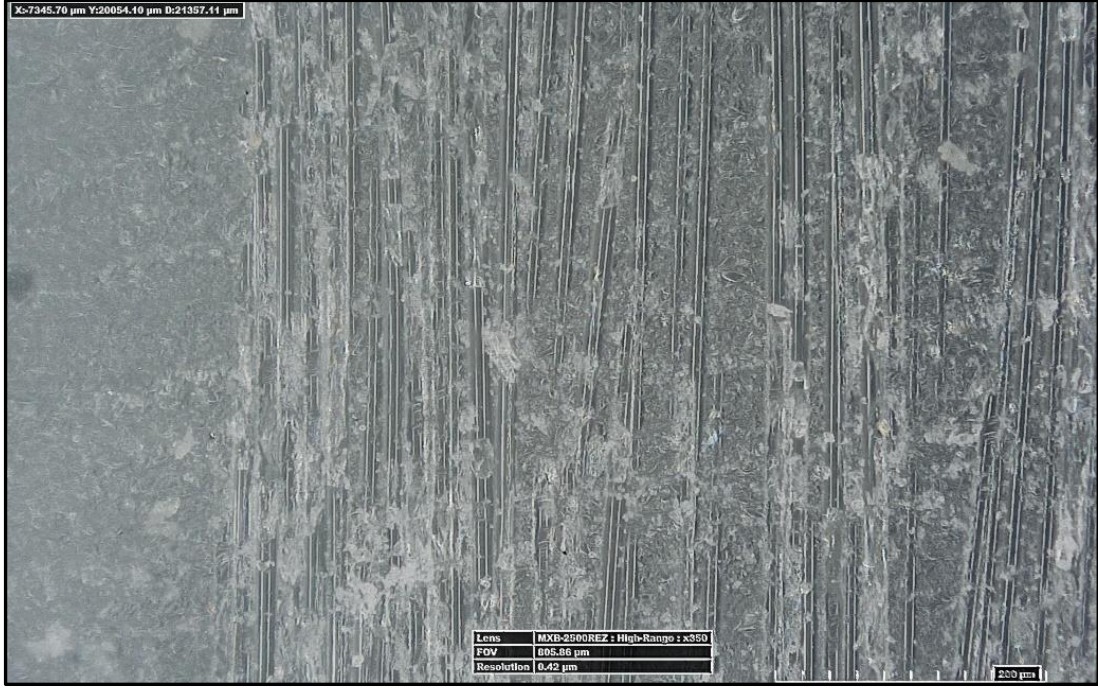

**Figure 7.** *Cont.*

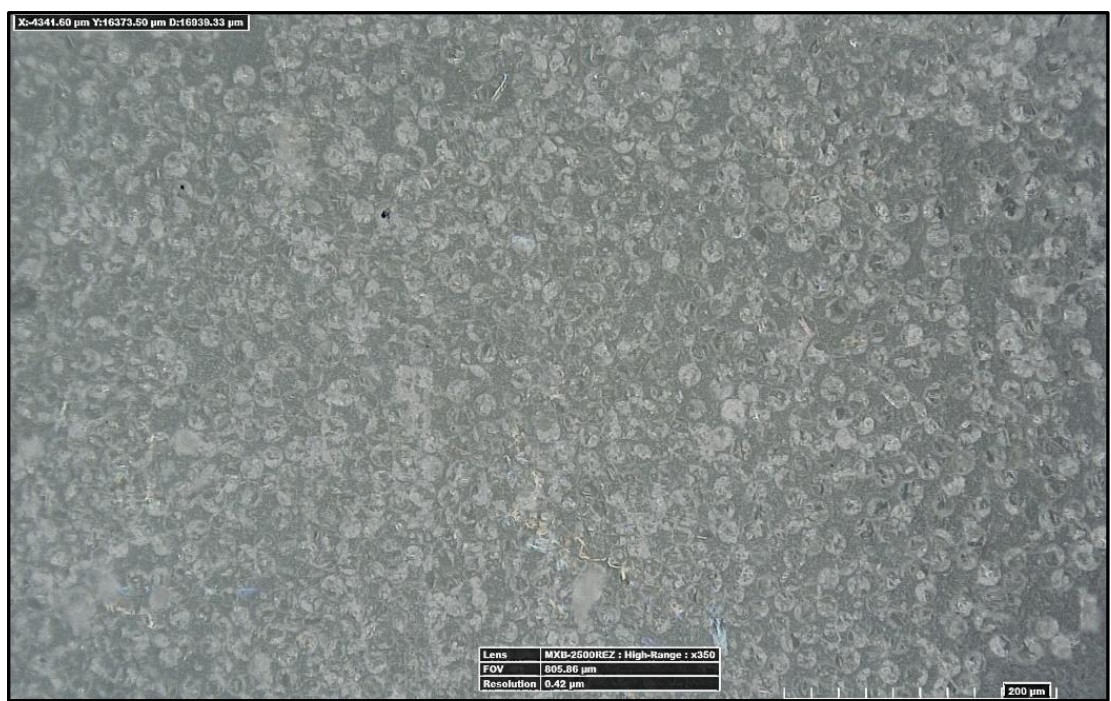

**Figure 7.** R-glass FRP plate with a (**bottom**; ×350 magnification) transverse and (**top**; ×350 magnification) hoop fiber orientation after a week of exposure to pH 10.0 aqueous solution.

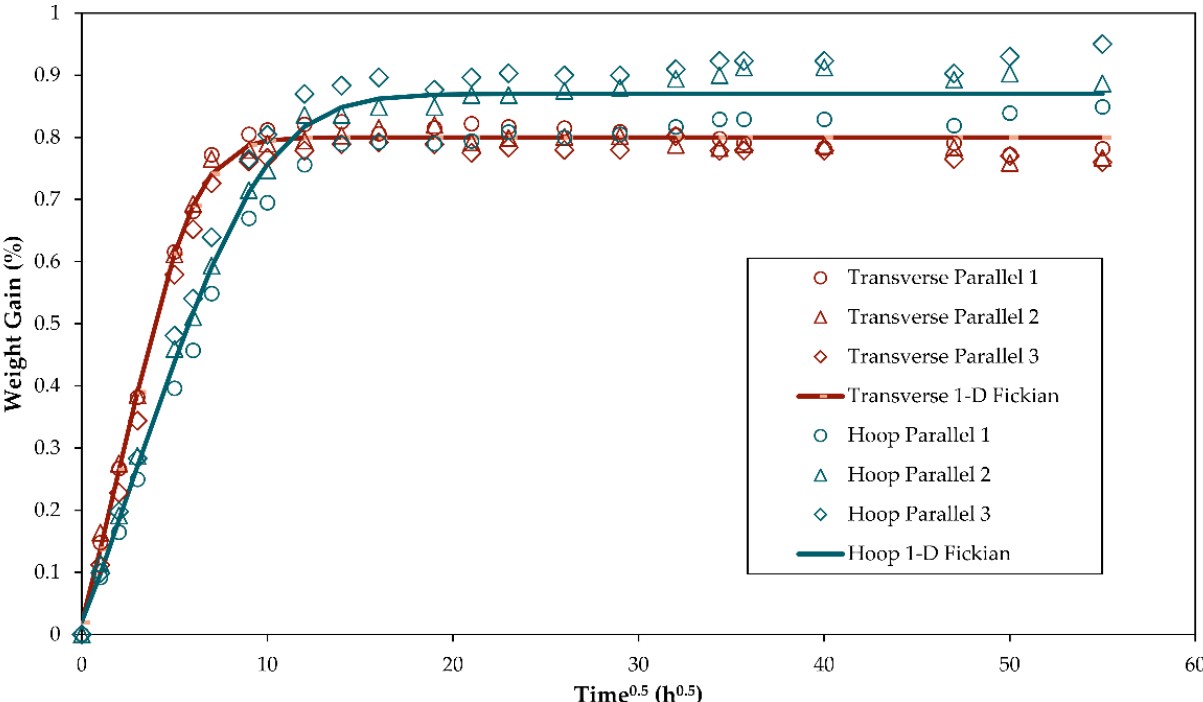

**Figure 8.** Water diffusion measurements.

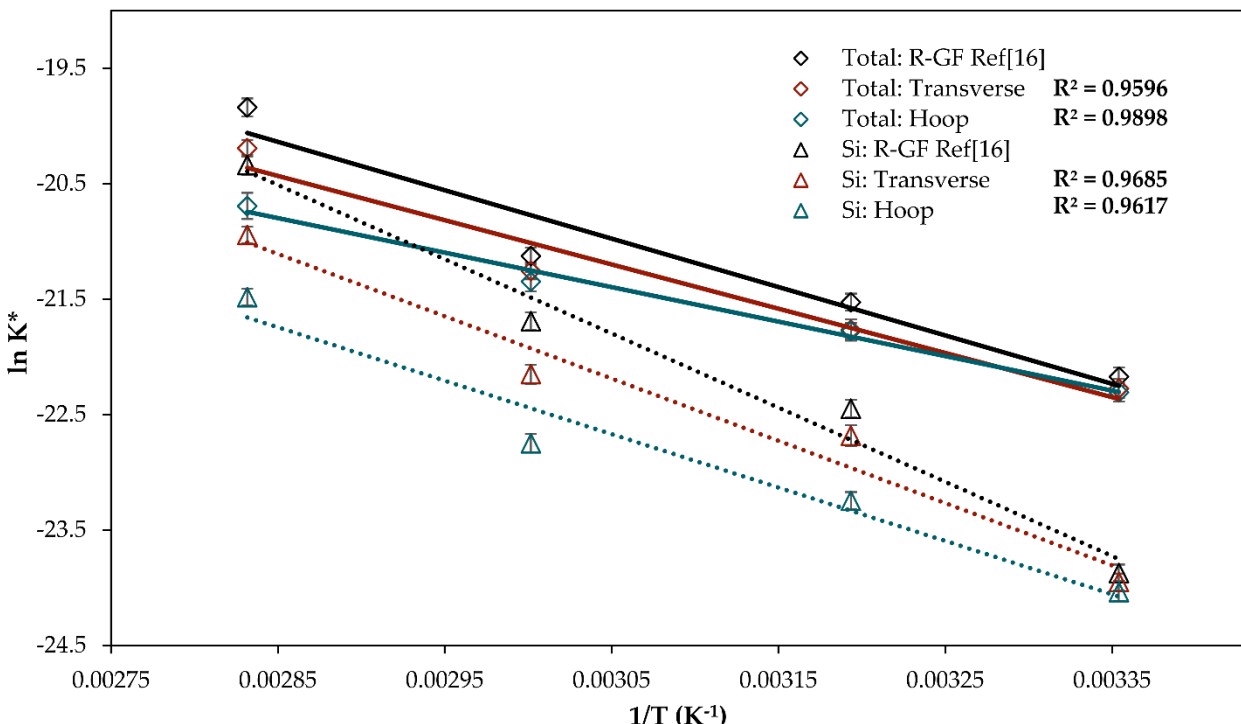

**Figure 9.** Graphing approach to obtain the pre-exponential factors and activation energies of Si and total glass dissolution for GFRPs at pH 5.65 and 60 °C compared with unencapsulated R-GF data reported in [16].

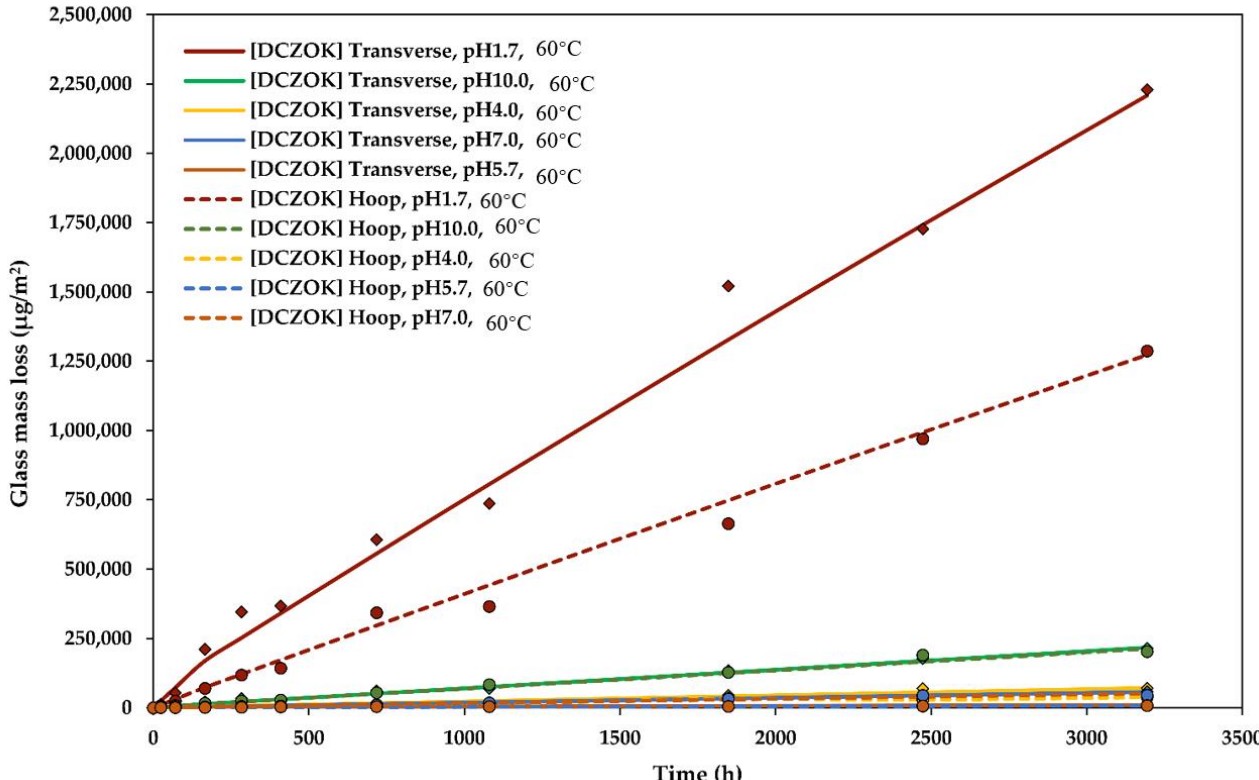

**Figure 10.** Glass dissolution kinetics at various pH levels: experimental data and modeled DCZOK dissolution curves for GFRP plates.

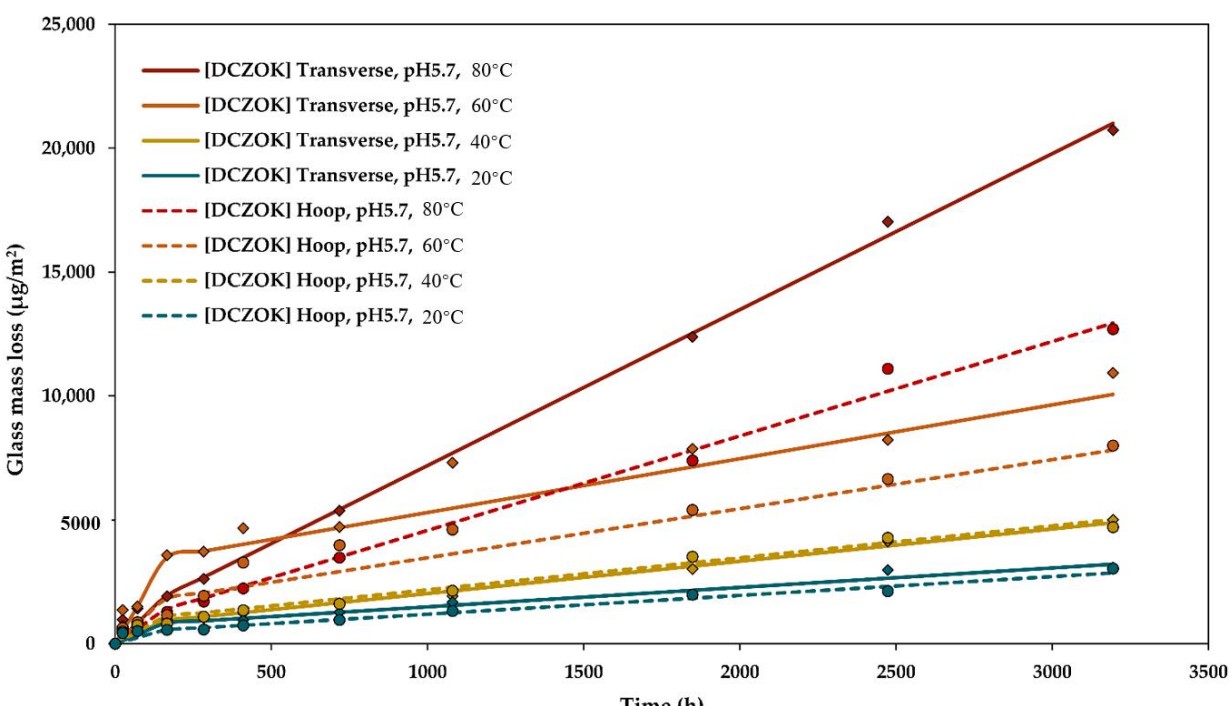

**Figure 11.** Glass dissolution kinetics at various temperatures: experimental data and modeled DCZOK dissolution curves for GFRP plates.

Even though the studied R-glass was boron-free (without any additional boron introduced during manufacturing, as indicated by the datasheets), it still degraded rapidly at pH 1.7 (Figures 3 and 4). For more acidic applications, the use of ECR-glass fiber-reinforced composites should be considered and evaluated, as they are likely more suitable than the R-glass composites [17], but experimental trials must be performed to verify this.

**Diffusion measurements.** The experimental gravimetric measurements for composite plates were performed to ensure that the R-GFRPs were saturated with water during the steady-state ion release kinetics measurements; the experimental weight gain curves are reported in Figure 8 for 4 months or 3000 h (54.7 $h^{0.5}$), along with simulated Fickian diffusion.

The modeling curves shown in Figure 8 were obtained using 1-D analytical solutions to Fick's second law applied to anisotropic materials described by Crank [53] and, more specifically, to the plate case, as reported in [54]. Relations for calculating water contents $w(t)$ were obtained by integration over the sample volume; these can be found elsewhere for orthotropic composite plates [54].

The equilibrium water uptake coefficients were 0.87 ± 0.08 wt. % and 0.80 ± 0.05 wt. % for hoop and transverse specimens, respectively. The water uptake was faster for composite plates with fibers in the transverse orientation than for those in the hoop orientation. This is in accord with the literature on water diffusion in plates [54,55]. The diffusivities were 0.0038 ± 0.0008 $mm^2$/h and 0.0108 ± 0.0005 $mm^2$/h for hoop and transverse specimens, respectively. The goodness-of-fit measures for hoop and transverse specimens were 0.9970 and 0.9993 (determined by $R^2$).

There was a long-term mass gain and mass loss observed for hoop and transverse specimens, respectively, which is in accord with previous findings on the hydrolytic flaw formation mechanism in the composite interphase, described in detail in [26].

**Ion release measurements.** Ion release was measured for 4 months (3000 h) as well. From the cumulative measured concentration data of released ions during the dissolution of glass, the dissolution rate constants were obtained using non-linear regression for Si ($K^I_{0_{Si}}$ and $K^{II}_{0_{Si}}$) and for the total mass loss ($K^I_{0_{total}}$ and $K^{II}_{0_{total}}$). The Generalized Reduced Gradient (GRG) non-linear regression method was used by minimizing the sum of squares

of the differences between modeled and experimental values. The obtained parameters are summarized in Tables 2 and 3 for the impact of pH and temperature on dissolution kinetics, respectively. The obtained values are reasonable (similar order of magnitude) when compared with the dissolution rates of other glass fibers studied in the literature [16]. The dissolution rate constants for glass dissolution from GFRPs in water at various pH values and $60 \pm 1\,^\circ$C are shown in Table 2. The steady state was achieved after about a week (about the same time as saturation with water).

**Table 2.** The glass dissolution rate constants were obtained via regression of the experimental data of GFRPs using the DCZOK model for Si and total mass loss at 60 $^\circ$C and various pH values for both hoop and transverse orientations.

| *pH* | $K_{0_{Si}}^{I}$ (g/(m$^2$·s)) | $K_{0_{Si}}^{II}$ (g/(m$^2$·s)) | $K_{0_{total}}^{I}$ (g/(m$^2$·s)) | $K_{0_{total}}^{II}$ (g/(m$^2$·s)) | *References* |
|---|---|---|---|---|---|
| **R-GF** | | | | | |
| $1.679 \pm 0.010$ | $(3.10 \pm 0.44) \times 10^{-7}$ | $(1.25 \pm 0.09) \times 10^{-7}$ | $(1.70 \pm 0.19) \times 10^{-6}$ | $(1.16 \pm 0.08) \times 10^{-6}$ | [16] |
| $4.005 \pm 0.010$ | $(2.59 \pm 0.33) \times 10^{-8}$ | $(1.70 \pm 0.11) \times 10^{-8}$ | $(8.48 \pm 1.21) \times 10^{-8}$ | $(6.24 \pm 0.36) \times 10^{-8}$ | [16] |
| $5.650 \pm 0.010$ | $(6.67 \pm 1.03) \times 10^{-9}$ | $(2.30 \pm 0.16) \times 10^{-9}$ | $(1.82 \pm 0.29) \times 10^{-8}$ | $(4.05 \pm 0.29) \times 10^{-9}$ | [16] |
| $7.000 \pm 0.010$ | $(3.64 \pm 0.53) \times 10^{-8}$ | $(2.55 \pm 0.19) \times 10^{-8}$ | $(5.46 \pm 0.82) \times 10^{-8}$ | $(4.85 \pm 0.38) \times 10^{-8}$ | [16] |
| $10.012 \pm 0.010$ | $(8.97 \pm 1.27) \times 10^{-8}$ | $(4.56 \pm 0.32) \times 10^{-8}$ | $(1.39 \pm 0.16) \times 10^{-7}$ | $(1.11 \pm 0.07) \times 10^{-7}$ | [16] |
| **Hoop R-GFRP** | | | | | |
| $1.679 \pm 0.010$ | $(2.58 \pm 0.44) \times 10^{-8}$ | $(1.80 \pm 0.12) \times 10^{-8}$ | $(1.19 \pm 0.18) \times 10^{-7}$ | $(1.10 \pm 0.08) \times 10^{-7}$ | This work |
| $4.005 \pm 0.010$ | $(8.80 \pm 1.27) \times 10^{-10}$ | $(8.12 \pm 0.69) \times 10^{-10}$ | $(4.60 \pm 0.69) \times 10^{-9}$ | $(3.50 \pm 0.32) \times 10^{-9}$ | This work |
| $5.650 \pm 0.010$ | $(1.92 \pm 0.32) \times 10^{-10}$ | $(1.32 \pm 0.10) \times 10^{-10}$ | $(2.95 \pm 0.41) \times 10^{-9}$ | $(5.35 \pm 0.42) \times 10^{-10}$ | This work |
| $7.000 \pm 0.010$ | $(9.00 \pm 1.10) \times 10^{-10}$ | $(8.20 \pm 0.53) \times 10^{-10}$ | $(5.32 \pm 0.71) \times 10^{-9}$ | $(4.28 \pm 0.28) \times 10^{-9}$ | This work |
| $10.012 \pm 0.010$ | $(1.47 \pm 0.25) \times 10^{-8}$ | $(7.44 \pm 0.71) \times 10^{-9}$ | $(2.28 \pm 0.35) \times 10^{-8}$ | $(1.78 \pm 0.11) \times 10^{-8}$ | This work |
| **Transverse R-GFRP** | | | | | |
| $1.679 \pm 0.010$ | $(4.98 \pm 0.72) \times 10^{-8}$ | $(2.03 \pm 0.12) \times 10^{-8}$ | $(2.76 \pm 0.42) \times 10^{-7}$ | $(1.91 \pm 0.13) \times 10^{-7}$ | This work |
| $4.005 \pm 0.010$ | $(1.80 \pm 0.29) \times 10^{-9}$ | $(1.70 \pm 0.11) \times 10^{-9}$ | $(7.00 \pm 1.08) \times 10^{-9}$ | $(6.00 \pm 0.47) \times 10^{-9}$ | This work |
| $5.650 \pm 0.010$ | $(7.71 \pm 0.92) \times 10^{-10}$ | $(2.40 \pm 0.19) \times 10^{-10}$ | $(5.66 \pm 0.79) \times 10^{-9}$ | $(5.87 \pm 0.41) \times 10^{-10}$ | This work |
| $7.000 \pm 0.010$ | $(2.00 \pm 0.24) \times 10^{-9}$ | $(1.90 \pm 0.12) \times 10^{-9}$ | $(5.80 \pm 0.78) \times 10^{-9}$ | $(4.80 \pm 0.35) \times 10^{-9}$ | This work |
| $10.012 \pm 0.010$ | $(1.46 \pm 0.21) \times 10^{-8}$ | $(7.51 \pm 0.53) \times 10^{-9}$ | $(2.30 \pm 0.33) \times 10^{-8}$ | $(1.82 \pm 0.14) \times 10^{-8}$ | This work |

The dissolution rate constants for glass dissolution from GFRPs in water at various temperatures and pH 5.65 are shown in Table 3. The steady state was also achieved after about a week.

The Arrhenius approach was used to obtain steady-state Si and glass dissolution's activation energy. The activation energy was obtained by graphing at constant *pH* and $\sigma$ (zero-stress conditions in this study), similar to what was implemented in previous studies (Equation (21)) [9,16]:

$$lnK_0 = -\frac{E_A}{R}\frac{1}{T} + lnA \tag{21}$$

The graphing approach is shown in Figure 9. The obtained apparent activation energy $E_A$ of Si dissolution was 38.44 and 44.89 kJ/mol (using $K_{0_{Si}}^{II}$ values) for hoop and transverse R-GFRPs, respectively, whereas for unembedded fibers, it was 53.46 kJ/mol [16]. The pre-exponential factor $A$ for Si dissolution was $6.82 \times 10^{-1}$ g/(m$^2$·s) for both configurations.

The obtained activation energy $E_A$ of total glass dissolution was 24.85 and 31.90 kJ/mol (using $K_{0_{total}}^{II}$ values) for hoop and transverse R-GFRPs, respectively. The pre-exponential factor for glass dissolution $A$ was $1.67 \times 10^{-3}$ g/(m$^2$·s) for all configurations. For the non-

encapsulated R-GF, the activation energy $E_A$ of total glass dissolution was 34.84 kJ/mol, which is the value reported in [16].

**Table 3.** The glass dissolution rate constants were obtained via regression of the experimental data of GFRPs using the DCZOK model for Si and total mass loss at pH 5.65 and various temperatures for both hoop and transverse orientations.

| $T$ (°C) | $K^I_{0_{Si}}$ (g/(m²·s)) | $K^{II}_{0_{Si}}$ (g/(m²·s)) | $K^I_{0_{total}}$ (g/(m²·s)) | $K^{II}_{0_{total}}$ (g/(m²·s)) | *References* |
|---|---|---|---|---|---|
| | | **R-GF** | | | |
| $20 \pm 1$ | $(1.46 \pm 0.23) \times 10^{-9}$ | $(2.60 \pm 0.18) \times 10^{-10}$ | $(1.04 \pm 0.12) \times 10^{-8}$ | $(1.42 \pm 0.11) \times 10^{-9}$ | [16] |
| $40 \pm 1$ | $(2.62 \pm 0.37) \times 10^{-9}$ | $(1.08 \pm 0.08) \times 10^{-9}$ | $(1.37 \pm 0.19) \times 10^{-8}$ | $(2.72 \pm 0.19) \times 10^{-9}$ | [16] |
| $60 \pm 1$ | $(6.67 \pm 1.03) \times 10^{-9}$ | $(2.30 \pm 0.16) \times 10^{-9}$ | $(1.82 \pm 0.29) \times 10^{-8}$ | $(4.05 \pm 0.29) \times 10^{-9}$ | [16] |
| $80 \pm 1$ | $(2.19 \pm 0.31) \times 10^{-8}$ | $(8.91 \pm 0.73) \times 10^{-9}$ | $(4.24 \pm 0.59) \times 10^{-8}$ | $(1.47 \pm 0.11) \times 10^{-8}$ | [16] |
| | | **Hoop R-GFRP** | | | |
| $20 \pm 1$ | $(6.96 \pm 0.97) \times 10^{-11}$ | $(3.64 \pm 0.28) \times 10^{-11}$ | $(9.02 \pm 0.12) \times 10^{-10}$ | $(2.05 \pm 0.16) \times 10^{-10}$ | This work |
| $40 \pm 1$ | $(1.71 \pm 0.22) \times 10^{-10}$ | $(8.02 \pm 0.58) \times 10^{-11}$ | $(1.78 \pm 0.27) \times 10^{-9}$ | $(3.49 \pm 0.23) \times 10^{-10}$ | This work |
| $60 \pm 1$ | $(1.92 \pm 0.32) \times 10^{-10}$ | $(1.32 \pm 0.10) \times 10^{-10}$ | $(2.95 \pm 0.41) \times 10^{-9}$ | $(5.35 \pm 0.42) \times 10^{-10}$ | This work |
| $80 \pm 1$ | $(5.78 \pm 0.91) \times 10^{-10}$ | $(4.68 \pm 0.33) \times 10^{-10}$ | $(2.26 \pm 0.35) \times 10^{-9}$ | $(1.03 \pm 0.11) \times 10^{-9}$ | This work |
| | | **Transverse R-GFRP** | | | |
| $20 \pm 1$ | $(2.97 \pm 0.42) \times 10^{-10}$ | $(3.97 \pm 0.28) \times 10^{-11}$ | $(1.35 \pm 0.19) \times 10^{-9}$ | $(2.12 \pm 0.17) \times 10^{-10}$ | This work |
| $40 \pm 1$ | $(3.46 \pm 0.48) \times 10^{-10}$ | $(1.41 \pm 0.12) \times 10^{-10}$ | $(1.52 \pm 0.21) \times 10^{-9}$ | $(3.52 \pm 0.31) \times 10^{-10}$ | This work |
| $60 \pm 1$ | $(7.71 \pm 0.92) \times 10^{-10}$ | $(2.40 \pm 0.19) \times 10^{-10}$ | $(5.66 \pm 0.79) \times 10^{-9}$ | $(5.87 \pm 0.41) \times 10^{-10}$ | This work |
| $80 \pm 1$ | $(1.10 \pm 0.22) \times 10^{-9}$ | $(8.02 \pm 0.56) \times 10^{-10}$ | $(3.15 \pm 0.42) \times 10^{-9}$ | $(1.70 \pm 0.12) \times 10^{-9}$ | This work |

The obtained values are consistent with values reported in the literature for R-GFs [16], but they are lower. This effect is considered apparent, which is due to the fact that ICP-MS captures only ions that have left the composite plates, whereas there is a significant number of ions accumulating inside. This issue is discussed in more detail in the Discussion section.

**Simulating the GFRP dissolution kinetics with the DCZOK model.** The ICP-MS data of ion release were used as the input for DCZOK model simulations. The results of the simulations are shown in Figures 10 and 11, where points represent experimental data, and lines represent DCZOK modeled curves using rate constants reported in Tables 2 and 3, respectively. In Figures 10 and 11, the glass mass loss is normalized per the initial surface area of fibers ($S_0$), as is commonly accepted for surface reaction kinetics [16]. $S_0$ was calculated based on geometrical considerations and the glass fiber fraction.

## 4. Discussion

**Effect of the environment.** As in the case of R-GFs [16], for the studied R-GFRPs, the temperature shows a similar Arrhenius-type influence on the kinetics, increasing the rate of dissolution exponentially with increasing temperature. The DCZOK model was able to capture the effect of temperature on R-GFRPs well in both hoop and transverse orientations.

Similar to R-GFs [16], R-GFRPs also showed a hyperbolic dependence on pH. Transverse specimens degraded much quicker at pH 1.7 than their hoop counterparts, and their integrity was destroyed within a week, as seen in Figure 4. Thus, R-GFRPs should not be used in strongly acidic conditions. This finding agrees with an observation in another study that stated that many GFRPs fail catastrophically after a critical time when exposed to acids [17,56]. It is possible to obtain the pH influence on the activation energies of dissolution by rearranging the Arrhenius equation into the following form (Equation (22)):

$$E_A = RT(lnA - lnK_0) \tag{22}$$

The pH function of the normalized activation energy of dissolution $E_A^{II} = f(pH)$ for R-glass [16] and the respective R-GFRPs studied in this work is shown in Figure 12. Normalization was performed with respect to activation energy at pH 5.65, which exhibited the lowest activation energy levels. It is commonly accepted that pH dependency can be approximated as a polynomial function [16,57].

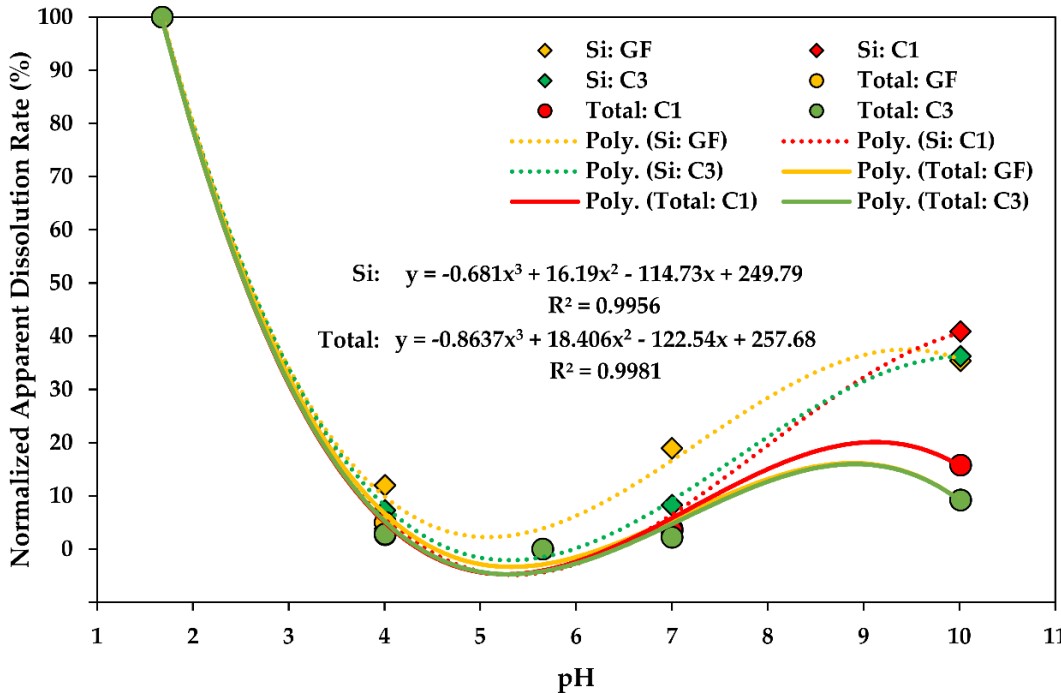

**Figure 12.** Normalized activation energy of the steady-state glass dissolution as a function of pH for R-GFs [16] and R-GFRPs (legend: C1 and C3 stand for hoop and transverse R-GFRP plates, respectively).

For all cases, the trend of the pH influence on the activation energy of dissolution is similar for both total glass dissolution and Si release. This observation indicates that the total glass dissolution rate constants for various pH levels form a master curve for glass fibers and the respective GFRPs. This is similarly true for Si release. Both glass and Si dissolution rate constant trends are similar, with the exception that at a high pH level, i.e., pH 10.0, the rate of Si dissolution increases more dramatically than the total glass dissolution rate, indicating a higher contribution of Si to the material loss during dissolution in basic environments.

**Effect of the fiber orientation and embedment.** Compared to the dissolution of un-embedded R-GFs [16], the long-term dissolution of the studied R-GFRPs was slowed down by 36.84% and 65.26% for R-GFRP plates with transverse and hoop fiber orientations, respectively. Slower dissolution from R-GFRPs compared to R-GFs is explained by the limited water availability and silica degradation product accumulation inside the composite.

**Effect of degradation product accumulation and water availability.** Unlike the free fibers, for GFRPs, the water availability and degradation product accumulation terms are essential and must be considered when studying the kinetics of glass degradation [9]. By using different sample configurations, the authors attempted to decouple the water availability and accumulation terms. By using a combination of short interface (sizing) highways (1.5 mm) in transverse R-GFRP plates, where accumulation can be assumed to be absent [9,10], and composite hoop R-GFRP plates with comparably long interface highways of 20 mm, it was possible to obtain the water availability and accumulation terms separately, similar to what was performed in earlier work [9,10]. The water availability is governed by the water saturation levels of the epoxy matrix $C_{H_2O}^{eq}$, which is 0.0344 g $H_2O$/g polymer (3.44 wt. %) for the studied polymer [9,50]. Unlike the hoop R-GFRP composite with long interface highways of 20 mm, for the 1.5 mm short-highway transverse R-GFRP

plates, it can be assumed that there is no accumulation term [10], so the time-dependent concentration of silica degradation products $C_{SiO_2}(t)$ is always 0. The whole accumulation term then equals 1 [10]. The DCZOK model simplifies to Equation (23):

$$\frac{\partial m}{\partial t} = 2n\pi l \left( r_0 K_0 \xi_{sizing} C_{H_2O}^{n_{order}}(t) \left( \frac{C_{SiO_2}^{eq}}{C_{SiO_2}^{eq}} \right)^{m_{order}} - \frac{\left( K_0 \xi_{sizing} C_{H_2O}^{n_{order}}(t) \left( \frac{C_{SiO_2}^{eq}}{C_{SiO_2}^{eq}} \right)^{m_{order}} \right)^2}{\rho_{glass}} t \right) \quad (23)$$

Since the composite plates are thin, the time-dependent water concentration in the polymeric matrix inside the composite $C_{H_2O}(t)$ reaches saturation $C_{H_2O}^{eq}$ very fast. Then, for steady-state dissolution, the equation simplifies to Equation (24):

$$\frac{\partial m}{\partial t} = 2n\pi l \left( r_0 K_0 \xi_{sizing} \left( C_{H_2O}^{eq} \right)^{n_{order}} - \frac{\left( K_0 \xi_{sizing} \left( C_{H_2O}^{eq} \right)^{n_{order}} \right)^2}{\rho_{glass}} t \right) \quad (24)$$

As can be seen in Equation (24), there are no more unknown time-dependent parameters left in the equation. The only unknown constant is the order of the water availability $n_{order}$, which then can be obtained from the experimental data using non-linear regression. Now, for the 20 mm long-highway composite hoop R-GFRP plates, the accumulation of the degradation products cannot be neglected [10], and thus, the mass loss formula in Equation (25) can be written as (after water saturation of the polymeric matrix):

$$\frac{\partial m}{\partial t} = 2n\pi l \left( r_0 K_0 \xi_{sizing} \left( C_{H_2O}^{eq} \right)^{n_{order}} \left( \frac{C_{SiO_2}^{eq} - C_{SiO_2}(t)}{C_{SiO_2}^{eq}} \right)^{m_{order}} - \frac{\left( K_0 \xi_{sizing} \left( C_{H_2O}^{eq} \right)^{n_{order}} \left( \frac{C_{SiO_2}^{eq} - C_{SiO_2}(t)}{C_{SiO_2}^{eq}} \right)^{m_{order}} \right)^2}{\rho_{glass}} t \right) \quad (25)$$

As can be seen in Equation (25), there are only two unknown parameters left in the equation, one being the time-dependent concentration of degradation products in the composite $C_{SiO_2}(t)$, and the other one being the constant order of the reaction $m_{order}$. Since $m_{order}$ is assumed to be the same for all conditions [10], $C_{SiO_2}(t)$ and $m_{order}$ are then obtained by fitting the model in Equation (25) to the experimental data. Both $C_{SiO_2}(t)$ and $C_{SiO_2}^{eq}$ are unitless (g $SiO_2$/g $H_2O \cdot$ g $H_2O$/g interphase) mass concentrations of silica degradation products per mass of water inside the composite interphase (Equation (26)):

$$C_{SiO_2}(t) = \frac{m_{accumulated\ SiO_2}(t)}{m_{interphase} \cdot C_{SiO_2}(t)} \cdot C_{SiO_2}(t) = \frac{m_{accumulated\ SiO_2}(t)}{m_{interphase}} \quad (26)$$

In order to calculate these values, it is necessary to calculate the volume and mass of the interphase $m_{interphase}$, which is assumed to be constant throughout the process. This can be performed on the basis of geometrical considerations [9]. The thickness of the interphase ($\delta_{interphase}$) is 0.05 μm, as reported in [9]. The volume is calculated as the difference between the outer (glass and interphase) and inner (bare glass) cylinders as follows (Equation (27)):

$$V_{interphase} = nl\pi \left( r_{sized\ fibres}^2 - r_{bare\ glass}^2 \right) \quad (27)$$

The density of the interphase is assumed to be the same as the density of the matrix polymer ($\rho_{interphase} = \rho_{matrix}$), since the interphase consists of about 90% epoxy film former by mass [9,58,59]. The weight of the interphase is roughly 1.01 wt. % of the sized fibers, which is consistent with another work (0.64 wt. % of the sized fibers determined via

burn-off tests) [9]. It is higher than in the case of burn-off tests since, in reality, the whole fiber surface is not uniformly covered with sizing [59]. One can then write Equation (28):

$$m_{interphase} \approx \frac{1.01\%}{100\%} m_{glass_{initial}} \tag{28}$$

where $m_{glass_{initial}}$ is the initial sized glass fiber mass (g).

The amount of water hosted by the interphase at saturation roughly corresponds to the water saturation levels of the studied epoxy matrix $C_{H_2O}^{eq}$, since about 90% of the interphase is the epoxy film former [9,58,59]. The equilibrium concentration of silica degradation products $C_{SiO_2}^{eq}$ is dependent on temperature and can be calculated using relationships provided in another work [9,16], which can be extended for the composite interphase (Equation (29)):

$$C_{SiO_2}^{eq} = -\frac{7.31 \times 10^{-4}}{T} C_{H_2O}^{eq} + 4.52 \times 10^{-6} \cdot C_{H_2O}^{eq} \tag{29}$$

Using Equation (18), the obtained saturation concentrations are 2.07, 2.19, 2.33, and 2.45 mg $SiO_2$/kg water, or $C_{SiO_2}^{eq}$ is equal to $7.11 \times 10^{-8}$, $7.52 \times 10^{-8}$, $8.00 \times 10^{-8}$, and $8.43 \times 10^{-8}$ g $SiO_2$/g interphase for 20, 40, 60, and 80 °C, respectively.

Measurements with different sample configurations allowed decoupling the water availability and degradation product accumulation terms at various pH levels and temperatures. Assuming that the effect of sizing $\xi_{sizing}$ is the same under all conditions and equals 0.165, as was reported in another study [10], the order of reaction $m_{order}$ was calculated for various environmental conditions at the steady state. The order of the reaction of the degradation product accumulation term was calculated for various conditions and is shown in Figure 13. The temperature did not exhibit a strong influence on the order of reaction for water availability.

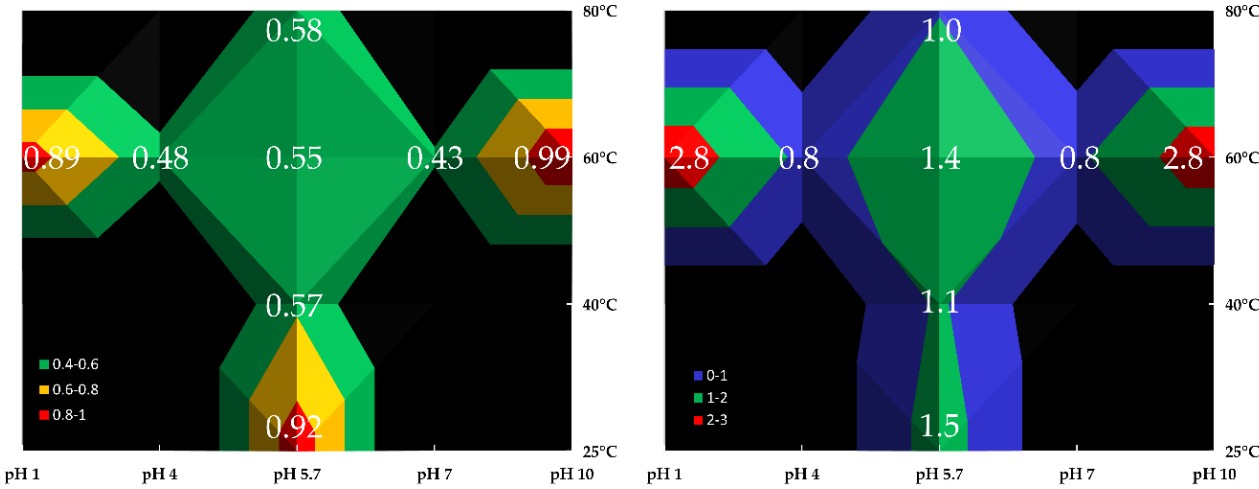

**Figure 13.** Effect of pH and temperature on the dissolution rate constant (**left**) and the dissolution activation energy (**right**). Black areas represent the absence of data.

The order of the water availability $n_{order}$ and the order of degradation product accumulation $m_{order}$ were obtained at steady-state dissolution, giving the best fit in regard to the experimental data. They were found to be 0.05 and 1, respectively. These values provided a good fit for all environmental conditions studied. A low value of $n_{order}$ indicates that the water is mostly in the free state, i.e., not chemically bound, which is consistent with another work on the same matrix polymer [9]. The obtained $m_{order}$ indicates that the accumulation reaction is linear and bears a close resemblance to Linear Driving Force (LDF) kinetics [60].

**Limitations of the DCZOK approach.** The authors think that experimentally and computationally validating the current study with the DCZOK model for seawater conditions (about 1.84–12.62 mg $SiO_2$/kg water; pH of seawater 7.8) would be highly beneficial,

especially for marine and offshore industries, since real-life structures most often operate in the seawater environment. When GFRPs are used in seawater, glass dissolution occurs more slowly due to the abundance of silica in seawater, which is due to seawater contact with sand and minerals [61,62]. The authors theorize that the model may account for this effect using the degradation product term since the degradation products $C_{SiO_2}(t)$ are already present in the seawater and should slow the degradation [61,62]. The approach in distilled water is thought to be conservative with regard to seawater, meaning that GFRP structures designed for distilled water conditions should not encounter penalties for their service time in the seawater.

Furthermore, the thickness of the composite is likely to have an influence. It should be more difficult for degradation products to leave a thicker composite. This would mean that the effect of degradation product accumulation should better protect thicker GFRP structures from glass dissolution. Additional research is needed to test this hypothesis. To use the model for thick structures, the water availability (water concentration) term should be found locally based on the diffusion profiles, i.e., using numerical Finite Element simulations, as described in another work [54]. The local time-dependent water concentrations can be obtained and operated locally in the analytical model by implementing the matrix polymer's water diffusion behavior.

The development of a more precise approach to determine the local degradation product concentration inside the composites is needed. This would allow predictions of the local deceleration of the reaction in thick structures and predictions of the time in which the degradation products reach saturation inside the composite if equilibrium can be achieved. It has been reported that, in some cases, after long water exposure, the deterioration of the strength of thick GFRP structures seems to stop, i.e., in some ship hulls, while in other cases, the deterioration proceeds, and saturation is not observed [63]. If equilibrium occurs, the authors believe this phenomenon may be linked to the accumulation of degradation products inside the structure. If this hypothesis is true, the dissolution of glass inside the composite should stop when $C_{SiO_2}^{eq}$ is reached. According to estimates using the DCZOK model with the parameters obtained in this work, the time to reach $C_{SiO_2}^{eq}$ in various environmental conditions is shown in Figure 14.

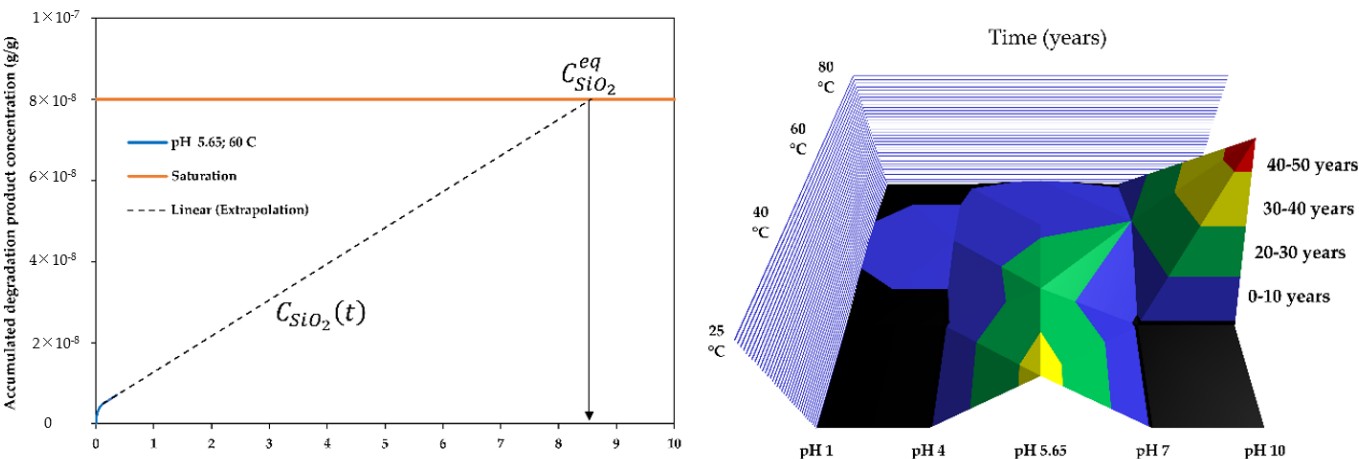

**Figure 14.** The extrapolation of the time-dependent degradation product concentrations in the interphase was used to estimate the time to saturation (**left**), and a summary of estimated times to reach the saturation of accumulated silica degradation products inside the composite for various environmental conditions is shown (**right**). Black areas represent the absence of data. Clarification: e.g., "8E3" stands for "$8 \times 10^3$".

The estimated times to reach the saturation of degradation products inside the composite and the approach used to obtain it are shown in Figure 14. The time to reach saturation decreases as the temperature is increased. At pH 5.65, the estimated time to reach saturation

at 20, 40, 60, and 80 °C is 24, 15, 8.5, and 1.5 years, respectively. Additionally, the time to saturate decreases as the environment becomes more acidic. At 60 °C, the estimated time to reach saturation at pH levels of 1.7, 4.0, 5.7, 7.0, and 10.0 is 1.5, 2.5, 8.5, 10, and 36 years, respectively.

In this work, the material dissolution is discussed only for glass fibers, but the matrix dissolution is omitted. It is important to add that matrix hydrolysis and dissolution can play a major role in mechanical property degradation. However, for the studied material, the dissolution of the matrix, along with the effect on mechanical properties, was studied previously, and it was shown that this particular matrix does not undergo hydrolysis [58,64] but is only affected by physical aging and plasticization [64], cosmetic yellowing [58], hygroscopic swelling [55], and viscoelastic creep [65].

Furthermore, the analysis of the mechanical properties over time is of great importance since GFRPs are usually heavily used in constructional applications [66–73]. Among these studies, only a few investigated the environmental aging-induced deterioration of the mechanical properties of the same R-GFRPs [69,72]. The mechanistic link between the dissolution kinetics of GFs and their strength loss kinetics was previously described in [16]. However, a clearer quantitative link between the loss of material in GFRPs and their strength deterioration is yet to be established. Therefore, relating degradation to the mechanics of the studied materials and their mechanical characteristics is seen as an important next step for the modeling of environmental aging of GFRPs.

**Types of GFRPs.** R-GFRPs were studied in this work. However, there should be no limitations to applying the DCZOK model to other types of glass fibers and GFRPs. The model should be applicable to other types of glass since SiO2 is the major component in virtually all types of glass [9], but it would be beneficial to validate this model experimentally with other types of glass fibers.

## 5. Conclusions

An analytical DCZOK model was applied to simulate the long-term glass dissolution experiments of R-GFRPs. The model accounts for the influence of pH, temperature, and the effects of sizing protection, the accumulation of degradation products inside the composite, and water availability. The glass dissolution rate constants were obtained and reported for various pH levels and temperatures.

Temperature exhibited an Arrhenius-type influence on the kinetics of R-GFRP dissolution, increasing the rate of dissolution exponentially with increasing temperature. The activation energy of steady-state glass dissolution was obtained and reported for composites with different fiber orientations (hoop and transverse). The effect was generally similar to that of the temperature effect on the unembedded R-GF.

The trend of pH influence on the activation energy of dissolution was similar and formed almost a master curve for R-GFs and the respective R-GFRPs in both hoop and transverse orientations. In comparison with neutral conditions, basic and acidic aqueous environments showed an increase in dissolution rates, negatively affecting the lifetime of glass fiber composites. A higher contribution of Si release to the material loss during dissolution in basic environments was observed. Composite samples quickly degraded in a strongly acidic environment and were destroyed within a week.

The slower dissolution from composites compared to fibers was due to the effects of limited water availability and due to degradation product accumulation inside the composite. The order of the degradation product accumulation term was theorized and compared for various pH levels and temperatures.

The slower dissolution from composites compared to fibers was due to the effects of limited water availability and due to degradation product accumulation inside the composite. The order of the water availability and the order of the degradation product accumulation term were obtained and were 0.05 and 1, respectively. A low value of $n_{order}$ indicates that the water is mostly in a free state, meaning primarily not chemically bound.

The obtained $m_{order}$ indicates that the degradation product accumulation is linear and resembles that of Linear Driving Force (LDF) kinetics.

**Author Contributions:** Conceptualization, A.E.K.; methodology, A.E.K., I.Z., I.B. and M.K.; software, A.E.K. and I.B.; validation, A.E.K.; formal analysis, A.E.K.; investigation, A.E.K.; resources, A.E.K. and M.K.; data curation, A.E.K., H.A.A., S.B., J.B., I.Z., I.B. and M.K.; writing—original draft preparation, A.E.K., H.A.A., J.B. and M.K.; writing—review and editing, A.E.K., H.A.A., S.B., J.B., I.Z., I.B. and M.K.; visualization, A.E.K.; supervision, A.E.K.; project administration, A.E.K.; funding acquisition, A.E.K. All authors have read and agreed to the published version of the manuscript.

**Funding:** This work was funded by the European Regional Development Fund within Activity 1.1.1.2 "Post-doctoral Research Aid" of Specific Aid Objective 1.1.1 of the Operational Programme "Growth and Employment" (Nr.1.1.1.2/VIAA/4/20/606, "Modelling Toolbox for Predicting Long-Term Performance of Structural Polymer Composites under Synergistic Environmental Ageing Conditions").

**Institutional Review Board Statement:** Not applicable.

**Informed Consent Statement:** Not applicable.

**Data Availability Statement:** Not applicable.

**Acknowledgments:** This work is part of a Postdoctoral Project research funded by the European Regional Development Fund within Activity 1.1.1.2 "Post-doctoral Research Aid" of the Specific Aid Objective 1.1.1 of the Operational Programme "Growth and Employment" (Nr.1.1.1.2/VIAA/4/20/606). The authors are thankful to Andreas T. Echtermeyer for fruitful discussions and Syverin Lierhagen and Konstantins Viligurs for help with elemental analyses. Andrey is grateful to Oksana.

**Conflicts of Interest:** The authors declare no conflict of interest. The funders had no role in the design of the study; in the collection, analyses, or interpretation of data; in the writing of the manuscript; or in the decision to publish the results.

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
