# Peer review of "Influence of Environmental Parameters and Fiber Orientation on Dissolution Kinetics of Glass Fibers in Polymer Composites"

_jcs, doi:10.3390/jcs6070210_

Round 1

Reviewer 1 Report

The authors applied Dissolving Cylinder Zero-Order Kinetic model to computationally study the long-term dissolution of glass composite materials. The model was verified by experiments on R-glass fiber reinforced composites. Overall, the paper is well prepared and organized, the study itself is adequate and clearly described.

My main concern is that the material dissolution is discussed only as for glass fibers, but the matrix dissolution is left behind. In my opinion, matrix dissolution can play a major role in mechanical properties degradation. Also, the analysis of the mechanical properties over time would be nice to see since GFRPs are usually are heavily used in constructional applications. 

Author Response

We want to express our sincere gratitude to the respected reviewer for the time dedicated to the review and the comprehensive, profound, and constructive remarks, which allowed us to improve the quality of our manuscript. The added or changed text of the manuscript was marked using “track changes” of Microsoft Word. We are very grateful for your positive outcome and for all your comments, as they have improved the text and helped to understand our study more properly.

We thank the reviewer for drawing our attention to the issue of matrix dissolution and effect on mechanical properties. The respected reviewer is absolutely right about the importance of matrix hydrolysis and dissolution. For our studied material, the dissolution of matrix along with the effect on mechanical properties was studied – and it was shown that this particular matrix does not undergo hydrolysis, see [REF]. We have addressed this comment in the text and added the information in the revised mansucript. The authors would like to thank the reviewer for this very valuable comment and addition to the paper.

Ref: Krauklis, Andrey E., Gagani, Abedin I. and Echtermeyer, Andreas T.. "Hygrothermal Aging of Amine Epoxy: Reversible Static and Fatigue Properties" Open Engineering, vol. 8, no. 1, 2018, pp. 447-454. https://doi.org/10.1515/eng-2018-0050

Yours sincerely,

Dr. Andrey E. Krauklis

Reviewer 2 Report

The present manuscript is a very good research work related to the effect of environmental conditions on the fiber orientation and dissolution of glass fibers in polymers composites.

The text is well written, well organized and very clear for the readers.

The experiments, methodology and the approaches of the authors are properly imposed. The literature review covers a wide and sound background, and worth highlighting they are well confronted to the research of the authors.

The article fuldill the criterias to be published and in my opinion will become a good contribution for the scientific community.

Author Response

We want to express our sincere gratitude to the respected reviewer for the time dedicated to the review and the comprehensive, profound, and constructive remarks, which allowed us to improve the quality of our manuscript. The table below presents in detail how to address each comment; the references are to the final line numbers of the revised article. In addition, the added or changed text of the manuscript was marked using “track changes” of Microsoft Word. We believe that, this paper will be cited frequently by other authors.

Comment of Reviewer #2

Response

The present manuscript is a very good research work related to the effect of environmental conditions on the fiber orientation and dissolution of glass fibers in polymers composites.

The text is well written, well organized and very clear for the readers.

We would like to take this opportunity to acknowledge the time and effort devoted by the reviewer.

We would like to thank the reviewer for this encouraging feedback.

The experiments, methodology and the approaches of the authors are properly imposed. The literature review covers a wide and sound background, and worth highlighting they are well confronted to the research of the authors.

We  thank the reviewer for this comment.

The article fuldill the criterias to be published and in my opinion will become a good contribution for the scientific community.

Again, many thanks are addressed to the respected reviewer for this positive feedback.

Yours sincerely,

Dr. Andrey E. Krauklis

Reviewer 3 Report

The paper seeks to introduce an approach ‘’ Influence of Environmental Parameters and Fiber Orientation on Dissolution Kinetics of Glass Fibers in Polymer Composites”. However, the authors should consider improving upon the quality to further highlight and emphasize. 

1.    In your abstract, consider adding one or two lines highlighting the significance of the study at the end.

2.    How were the samples prevented from being contaminated during the period of drying for the FT-IR?

3.    The introduction needs to be improved by relating to the mechanics of the studied materials and their mechanical characteristics. The references to be included are: 10.1007/s10853-022-06994-3, 10.3390/polym14132662, 10.1016/j.jiec.2022.06.023, 10.1016/j.polymertesting.2017.09.009, and 10.1016/j.compstruct.2021.114698.

4.    In lines 266 and 268, change the word “was” to “were”.

5.    In line 279, you stated that “The specimens were placed in inert closed vessels filled with 50 mL of distilled water or pH buffer solutions”. What happens if it’s kept in only the distilled water but not the pH and vice versa. If it’s the same for both. What accounts for that? And what is the reason for keeping the samples there?

6.    The magnification footers in the various figure especially figure 5 is not visible due to smaller font size. Manually add them to the figures for proper identification. Also, unify texts for all figures making sure they do not appear blur as some of the are currently showing.

7.    What was the accelerating voltage, scale bar, and the working range applied during the microscopy?

8.    Full stop is put in front of any abbreviated unit and space before introducing another unit. For instance, in line 436, these were written as “0.87 ± 0.08 wt% and 0.80 ± 0.05 wt%” instead of “0.87 ± 0.08 wt. % and 0.80 ± 0.05 wt. %”. Modify it accordingly.

9.    Put space between each variable and its corresponding unit. Instead of 36.84% and 65.26% in line 532, consider typing it as 36.84 % and 65.26 %.

Author Response

We want to express our sincere gratitude to the respected reviewer for the time dedicated to the review and the comprehensive, profound, and constructive remarks, which allowed us to improve the quality of our manuscript. The table below presents in detail how to address each comment. In addition, the added or changed text of the manuscript was marked using “track changes”.

Comment of Reviewer #3

Response

The paper seeks to introduce an approach ‘’ Influence of Environmental Parameters and Fiber Orientation on Dissolution Kinetics of Glass Fibers in Polymer Composites”. However, the authors should consider improving upon the quality to further highlight and emphasize. 

1.    1. In your abstract, consider adding one or two lines highlighting the significance of the study at the end.

We are very grateful for your positive outcome and for all your comments, as they have improved the text and helped to understand our study more properly. Please find below our responses to each of your comments. We have expanded the abstract with your suggestion about the significance. Thank you! The following was added: ‘The significance of the study is to contribute to a better understanding of mass loss and dissolution modelling in GFRPs, which is linked to deterioration of mechanical properties of GFRPs. This link should be further investigated experimentally and computationally.’

2. How were the samples prevented from being contaminated during the period of drying for the FT-IR?

Thank you for this remark.  The specimens were carefully handled during the whole procedure as described in other works, where the respective FTIR method was developed and implemented, refs [9,50].

3. The introduction needs to be improved by relating to the mechanics of the studied materials and their mechanical characteristics. The references to be included are: 10.1007/s10853-022-06994-3, 10.3390/polym14132662, 10.1016/j.jiec.2022.06.023, 10.1016/j.polymertesting.2017.09.009, and 10.1016/j.compstruct.2021.114698.

Thank you very much for drawing our attention to these issues. Consequently, we have added all of the missing references you have kindly provided: 10.1007/s10853-022-06994-3

10.3390/polym14132662

10.1016/j.jiec.2022.06.023

10.1016/j.polymertesting.2017.09.009

10.1016/j.compstruct.2021.114698, and a few more on similar materials, including the same R-GFRPs. However, we decided to add this in the discussion rather than the introduction, since the discussion on the influence on mechanical properties fits better in the Discussion, as the continuation of this study is seen as the link to mechanical deterioration, but it is indeed a continuation of the kinetics study. We hope that the reviewer agrees with our decision to do so. Thank you kindly!

4. In lines 266 and 268, change the word “was” to “were”.

We thank the reviewer for this suggestion. Corrected as suggested.

5. In line 279, you stated that “The specimens were placed in inert closed vessels filled with 50 mL of distilled water or pH buffer solutions”. What happens if it’s kept in only the distilled water but not the pH and vice versa. If it’s the same for both. What accounts for that? And what is the reason for keeping the samples there?

We would like to thank the referee for bringing this issue to our attention. The question is very interesting. Some specimens were conditioned in pure distilled water, whereas others were conditioned in pH buffer solutions. The difference in aging kinetics therefore can be seen in the Results section. The reason for keeping the samples in these vessels is to ensure that none of the contaminants can come in contact with the specimens during the study. The procedure has shown to be effective, as described in the other paper for influence of pH on pure R-glass fibers [Ref 9 and 16]. Thank you for a very interesting comment.

6. The magnification footers in the various figure especially figure 5 is not visible due to smaller font size. Manually add them to the figures for proper identification. Also, unify texts for all figures making sure they do not appear blur as some of the are currently showing.

Thank you for this comment. We do not have a better quality microscop figures, unfortunately. However, we have addressed your comment by providing magnification information in the legend of the figures. Also, better quality figures were provided, where they were available throughout the text.

7. What was the accelerating voltage, scale bar, and the working range applied during the microscopy?

We thank the reviewer for this remark. However, the microscope used in this work was not an electron scanning microscope SEM. Therefore, there was no accelerating voltage. Regarding the magnification, the new information was provided in Figures 4-7.

8. Full stop is put in front of any abbreviated unit and space before introducing another unit. For instance, in line 436, these were written as “0.87 ± 0.08 wt% and 0.80 ± 0.05 wt%” instead of “0.87 ± 0.08 wt. % and 0.80 ± 0.05 wt. %”. Modify it accordingly.

We are very grateful for your comments. It was corrected as suggested by the reviewer.

 9. Put space between each variable and its corresponding unit. Instead of 36.84% and 65.26% in line 532, consider typing it as 36.84 % and 65.26 %.

Fixed as suggested by the respected reviewer.

Again, thank you for your suggestions that improved the text and the overall level of the manuscript.

Yours sincerely,

Dr. Andrey E. Krauklis

Round 2

Reviewer 3 Report

The authors have finalised the comments perfectly. The paper is accepted and can be published.